# Unsupervised Learning of Goal Spaces for Intrinsically Motivated Goal Exploration

**Alexandre Péré**
Flowers Team
Inria and Ensta-ParisTech, France
alexandre.pere@inria.fr

**Sebastien Forestier**
Flowers Team
Inria and Ensta-ParisTech, France
sebastien.forestier@inria.fr

**Olivier Sigaud**
Flowers Team
Inria, Ensta-ParisTech and UPMC, France
Olivier.Sigaud@upmc.fr

**Pierre-Yves Oudeyer**
Flowers Team
Inria and Ensta-ParisTech, France
pierre-yves.oudeyer@inria.fr

## Abstract

Intrinsically motivated goal exploration algorithms enable machines to discover repertoires of policies that produce a diversity of effects in complex environments. These exploration algorithms have been shown to allow real world robots to acquire skills such as tool use in high-dimensional continuous state and action spaces. However, they have so far assumed that self-generated goals are sampled in a specifically engineered feature space, limiting their autonomy. In this work, we propose to use deep representation learning algorithms to learn an adequate goal space. This is a developmental 2-stage approach: first, in a perceptual learning stage, deep learning algorithms use passive raw sensor observations of world changes to learn a corresponding latent space; then goal exploration happens in a second stage by sampling goals in this latent space. We present experiments where a simulated robot arm interacts with an object, and we show that exploration algorithms using such learned representations can match the performance obtained using engineered representations.
**Keywords: exploration; autonomous goal setting; diversity; unsupervised learning; deep neural network**

## 1 Introduction

Spontaneous exploration plays a key role in the development of knowledge and skills in human children. For example, young children spend a large amount of time exploring what they can do with their body and external objects, independently of external objectives such as finding food or following instructions from adults. Such intrinsically motivated exploration (Berlyne, 1966; Gopnik et al., 1999; Oudeyer & Smith, 2016) leads them to make ratcheting discoveries, such as learning to locomote or climb in various styles and on various surfaces, or learning to stack and use objects as tools. Equipping machines with similar intrinsically motivated exploration capabilities should also be an essential dimension for lifelong open-ended learning and artificial intelligence.

In the last two decades, several families of computational models have both contributed to a better understanding of such exploration processes in infants, and how to apply them efficiently for autonomous lifelong machine learning (Oudeyer et al., 2016). One general approach taken by several research groups (Baldassarre et al., 2013; Oudeyer et al., 2007; Barto, 2013; Friston et al., 2017) has been to model the child as intrinsically motivated to make sense of the world, exploring like a scientist that imagines, selects and runs experiments to gain knowledge and control over the world. These models have focused in particular on three kinds of mechanisms argued to be essential and complementary to enable machines and animals to efficiently explore and discover skill repertoires in the real world (Oudeyer et al., 2013; Cangelosi et al., 2015): embodiment [1], intrinsic motivation[2]

---

[1]Body synergies provide structure on action and perception
[2]Self-organizes a curriculum of exploration and learning at multiple levels of abstraction

and social guidance[3]. This article focuses on challenges related to learning goal representations for intrinsically motivated exploration, but also leverages models of embodiment, through the use of parameterized Dynamic Movement Primitives controllers (Ijspeert et al., 2013) and social guidance, through the use of observations of another agent.

Given an embodiment, intrinsically motivated exploration[4] consists in automatically and spontaneously conducting experiments with the body to discover both the world dynamics and how it can be controlled through actions. Computational models have framed intrinsic motivation as a family of mechanisms that self-organize agents exploration curriculum, in particular through generating and selecting experiments that maximize measures such as novelty (Andreae & Andreae, 1978; Sutton, 1990), predictive information gain (Little & Sommer, 2013), learning progress (Schmidhuber, 1991; Kaplan & Oudeyer, 2003), compression progress (Schmidhuber, 2013), competence progress (Baranes & Oudeyer, 2013), predictive information (Martius et al., 2013) or empowerment (Salge et al., 2014). When used in the Reinforcement Learning (RL) framework (e.g. (Sutton, 1990; Schmidhuber, 1991; Kaplan & Oudeyer, 2003; Barto, 2013)), these measures have been called intrinsic rewards, and they are often applied to reward the "interestingness" of actions or states that are explored. They have been consistently shown to enable artificial agents or robots to make discoveries and solve problems that would have been difficult to learn using a classical optimization or RL approach based only on the target reward (which is often rare or deceptive) (Chentanez et al., 2005; Baranes & Oudeyer, 2013; Stanley & Lehman, 2015). Recently, they have been similarly used to guide exploration in difficult deep RL problems with sparse rewards, e.g. (Bellemare et al., 2016; Houthooft et al., 2016; Tang et al., 2017; Pathak et al., 2017).

However, many of these computational approaches have considered intrinsically motivated exploration at the level of micro-actions and states (e.g. considering low-level actions and pixel level perception). Yet, children's intrinsically motivated exploration leverages abstractions of the environments, such as objects and qualitative properties of the way they may move or sound, and explore by setting self-generated goals (Von Hofsten, 2004), ranging from objects to be reached, toy towers to be built, or paper planes to be flown. A computational framework proposed to address this higher-level form of exploration has been Intrinsically Motivated Goal Exploration Processes (IMGEPs) (Baranes & Oudeyer, 2009; Forestier et al., 2017), which is closely related to the idea of goal babbling (Rolf et al., 2010). Within this approach, agents are equipped with a mechanism enabling them to sample a goal in a space of parameterized goals[5], before they try to reach it by executing an experiment. Each time they sample a goal, they dedicate a certain budget of experiments time to improve the solution to reach this goal, using lower-level optimization or RL methods for example. Most importantly, in the same time, they take advantage of information gathered during this exploration to discover other outcomes and improve solutions to other goals[6].

This property of cross-goal learning often enables efficient exploration even if goals are sampled randomly (Baranes & Oudeyer, 2013) in goal spaces containing many unachievable goals. Indeed, generating random goals (including unachievable ones) will very often produce goals that are outside the convex hull of already discovered outcomes, which in turn leads to exploration of variants of known corresponding policies, pushing the convex hull further. Thus, this fosters exploration of policies that have a high probability to produce novel outcomes without the need to explicitly measure novelty. This explains why forms of random goal exploration are a form of intrinsically motivated exploration. However, more powerful goal sampling strategies exist. A particular one consists in using meta-learning algorithms to monitor the evolution of competences over the space of goals and to select the next goal to try, according to the expected competence progress resulting from practicing it (Baranes & Oudeyer, 2013). This enables to automate curriculum sequences of goals of progressively increasing complexity, which has been shown to allow high-dimensional real world robots to acquire efficiently repertoires of locomotion skills or soft object manipulation (Baranes & Oudeyer, 2013), or advanced forms of nested tool use (Forestier et al., 2017). Similar ideas have been recently applied in the context of multi-goal deep RL, where architectures closely related to intrinsically motivated goal exploration are used by procedurally generating goals and

---

[3]Leverages what others already know

[4]Also called curiosity-driven exploration

[5]Here a goal is not necessarily an end state to be reached, but can characterize certain parameterized properties of changes of the world, such as following a parameterized trajectory.

[6]E.g. while learning how to move an object to the right, they may discover how to move it to the left.

sampling them randomly (Cabi et al., 2017; Najnin & Banerjee, 2017) or adaptively (Florensa et al., 2017).

Yet, a current limit of existing algorithms within the family of Intrinsically Motivated Goal Exploration Processes is that they have assumed that the designer[7] provides a representation allowing the autonomous agent to generate goals, together with formal tools used to measure the achievement of these goals (e.g. cost functions). For example, the designer could provide a representation that enables the agent to imagine goals as potential continuous target trajectories of objects (Forestier et al., 2017), or reach an end-state starting from various initial states defined in Euclidean space (Florensa et al., 2017), or realize one of several discrete relative configurations of objects (Cabi et al., 2017), which are high-level abstractions from the pixels. While this has allowed to show the power of intrinsically motivated goal exploration architectures, designing IMGEPs that sample goals from a learned goal representation remains an open question. There are several difficulties. One concerns the question of how an agent can learn in an unsupervised manner a representation for hypothetical goals that are relevant to their world before knowing whether and how it is possible to achieve them with the agent's own action system. Another challenge is how to sample "interesting" goals using a learned goal representation, in order to remain in regions of the learned goal parameters that are not too exotic from the underlying physical possibilities of the world. Finally, a third challenge consists in understanding which properties of unsupervised representation learning methods enable an efficient use within an IMGEP architecture so as to lead to efficient discovery of controllable effects in the environment.

In this paper, we present one possible approach named IMGEP-UGL where aspects of these difficulties are addressed within a 2-stage developmental approach, combining deep representation learning and goal exploration processes:

**Unsupervised Goal space Learning stage (UGL):** In the first phase, we assume the learner can passively observe a distribution of world changes (e.g. different ways in which objects can move), perceived through raw sensors (e.g. camera pixels or other forms of low-level sensors in other modalities). Then, an unsupervised representation learning algorithm is used to learn a lower-dimensional latent space representation (also called embedding) of these world configurations. After training, a Kernel Density Estimator (KDE) is used to estimate the distribution of these observations in the latent space.

**Intrinsically Motivated Goal Exploration Process stage (IMGEP):** In the second phase, the embedding representation and the corresponding density estimation learned during the first stage are reused in a standard IMGEP. Here, goals are iteratively sampled in the embedding as target outcomes. Each time a goal is sampled, the current knowledge (forward model and meta-policy, see below) enables to guess the parameters of a corresponding policy, used to initialize a time-bounded optimization process to improve the cost of this policy for this goal. Crucially, each time a policy is executed, the observed outcome is not only used to improve knowledge for the currently selected goal, but for all goals in the embedding. This process enables the learner to incrementally discover new policy parameters and their associated outcomes, and aims at learning a repertoire of policies that produce a maximally diverse set of outcomes.

A potential limit of this approach, as it is implemented and studied in this article, is that representations learned in the first stage are frozen and do not evolve in the second stage. However, we consider here this decomposition for two reasons. First, it corresponds to a well-known developmental progression in infant development: in their first few weeks, motor exploration in infants is very limited (due to multiple factors), while they spend a considerable amount of time observing what is happening in the outside world with their eyes (e.g. observing images of social peers producing varieties of effects on objects). During this phase, a lot of perceptual learning happens, and this is reused later on for motor learning (infant perceptual development often happens ahead of motor development in several important ways). Here, passive perceptual learning from a database of visual effects observed in the world in the first phase can be seen as a model of this stage where infants learn by passively observing what is happening around them[8]. A second reason for this decomposi-

---

[7]Here we consider the human designer that crafts the autonomous agent system.

[8]Here, we do not assume that the learner actually knows that these observed world changes are caused by another agent, and we do not assume it can perceive or infer the action program of the other agent. Other works

tion is methodological: given the complexity of the underlying algorithmic components, analyzing the dynamics of the architecture is facilitated when one decomposes learning in these two phases (representation learning, then exploration).

**Main contribution of this article.** Prior to this work, and to our knowledge, all existing goal exploration process architectures used a goal space representation that was hand designed by the engineer, limiting the autonomy of the system. Here, the main contribution is to show that representation learning algorithms can discover goal spaces that lead to exploration dynamics close to the one obtained using an engineered goal representation space. The proposed algorithmic architecture is tested in two environments where a simulated robot learns to discover how to move and rotate an object with its arm to various places (the object scene being perceived as a raw pixel map). The objective measure we consider, called KL-coverage, characterizes the diversity of discovered outcomes during exploration by comparing their distribution with the uniform distribution over the space of outcomes that are physically possible (which is unknown to the learner). We even show that the use of particular representation learning algorithms such as VAEs in the IMGEP-UGL architecture can produce exploration dynamics that match the one using engineered representations.

**Secondary contributions of this article:**

- We show that the IMGEP-UGL architecture can be successfully implemented (in terms of exploration efficiency) using various unsupervised learning algorithms for the goal space learning component: AutoEncoders (AEs) (Bourlard & Kamp, 1988), Variational AE (VAE) (Rezende et al., 2014; Kingma & Ba, 2015), VAE with Normalizing Flow (Rezende & Mohamed, 2015), Isomap (Tenenbaum et al., 2000), PCA (Pearson, 1901), and we quantitatively compare their performances in terms of exploration dynamics of the associated IMGEP-UGL architecture.
- We show that specifying more embedding dimensions than needed to capture the phenomenon manifold does not deteriorate the performance of these unsupervised learning algorithms.
- We show examples of unsupervised learning algorithms (Radial Flow VAEs) which produce less efficient exploration dynamics than other algorithms in our experiments, and suggest hypotheses to explain this difference.

## 2 GOALS REPRESENTATION LEARNING FOR EXPLORATION ALGORITHMS

In this section, we first present an outline of intrinsically motivated goal exploration algorithmic architectures (IMGEPs) as originally developed and used in the field of developmental robotics, and where goal spaces are typically hand crafted. Then, we present a new version of this architecture (IMGEP-UGL) that includes a first phase of passive perceptual learning where goal spaces are learned using a combination of representation learning and density estimation. Finally, we outline a list of representation learning algorithms that can be used in this first phase, as done in the experimental section.

### 2.1 INTRINSICALLY MOTIVATED GOAL EXPLORATION ALGORITHMS

Intrinsically Motivated Goal Exploration Processes (IMGEPs), are powerful algorithmic architectures which were initially introduced in Baranes & Oudeyer (2009) and formalized in Forestier et al. (2017). They can be used as heuristics to drive the exploration of high-dimensional continuous action spaces so as to learn forward and inverse control models in difficult robotic problems. To clearly understand the essence of IMGEPs, we must envision the robotic agent as an experimenter seeking information about an unknown physical phenomenon through sequential experiments. In this perspective, the main elements of an exploration process are:

- A *context* $c$, element of a Context Space $\mathcal{C}$. This context represents the initial experimental factors that are not under the robotic agent control. In most cases, the context is considered fully observable (e.g. state of the world as measured by sensors).

---

have considered how stronger forms of social guidance, such as imitation learning (Schaal et al., 2003), could accelerate intrinsically motivated goal exploration (Nguyen & Oudeyer, 2014), but they did not consider the challenge of learning goal representations.

- A *parameterization* $\theta$, element of a Parameterization Space $\Theta$. This parameterization represents the experimental factors that can be controlled by the robotic agent (e.g. parameters of a policy).
- An *outcome* $o$, element of an Outcome Space $\mathcal{O}$. The outcome contains information qualifying properties of the phenomenon during the execution of the experiment (e.g. measures characterizing the trajectory of raw sensor observations during the experiment).
- A *phenomenon dynamics* $D : \mathcal{C}, \Theta \mapsto \mathcal{O}$, which in most interesting cases is unknown.

If we take the example of the *Arm-Ball* problem[9] in which a multi-joint robotic arm can interact with a ball, the context could be the initial state of the robot and the ball, the parameterization could be the parameters of a policy that generate a sequence of motor torque commands for $N$ time steps, and the outcome could be the position of the ball at the last time step. Developmental roboticists are interested in developing autonomous agents that learn two models, the forward model $\tilde{D} : \mathcal{C} \times \Theta \mapsto \mathcal{O}$ which approximates the phenomenon dynamics, and the inverse model $\tilde{I} : \mathcal{C} \times \mathcal{O} \mapsto \Theta$ which allows to produce desired outcomes under given context by properly setting the parameterization. Using the aforementioned elements, one could imagine a simple strategy that would allow the agent to gather tuples $\{c, \theta, o\}$ to train those models, by uniformly sampling a random parameterization $\theta \sim \mathcal{U}(\theta)$ and executing the experiment. We refer to this strategy as *Random Parameterization Exploration*. The problem for most interesting applications in robotics, is that only a small subspace of $\Theta$ is likely to produce interesting outcomes. Indeed, considering again the Arm-Ball problem with time-bounded action sequences as parameterizations, very few of those will lead the arm to touch the object and move it. In this case, a random sampling in $\Theta$ would be a terrible strategy to yield interesting samples allowing to learn useful forward and inverse models for moving the ball.

To overcome this difficulty, one must come up with a better approach to sample parameterizations that lead to informative samples. Intrinsically Motivated Goal Exploration Strategies propose a way to address this issue by giving the agent a set of tools to handle this situation:

- A *Goal Space* $\mathcal{T}$ whose elements $\tau$ represent parameterized goals that can be targeted by the autonomous agent. In the context of this article, and of the IMGEP-UGL architecture, we consider the simple but important case where the *Goal Space* is equated with the *Outcome space*. Thus, goals are simply vectors in the outcome space that describe target properties of the phenomenon that the learner tries to achieve through actions.
- A *Goal Policy* $\gamma(\tau)$, which is a probability distribution over the Goal Space used for sampling goals (see Algorithmic Architecture 2). It can be stationary, but in most cases, it will be updated over time following an intrinsic motivation strategy. Note that in some cases, this Goal Policy can be conditioned on the context $\gamma(\tau|c)$.
- A set of *Goal-parameterized Cost Functions* $C_\tau : \mathcal{O} \mapsto \mathbb{R}$ defined over all $\mathcal{O}$, which maps every outcome with a real number representing the goodness-of-fit of the outcome $o$ regarding the goal $\tau$. As these cost functions are defined over $\mathcal{O}$, this enables to compute the cost of a policy for a given goal even if the goal is imagined after the policy roll-out. Thus, as IMGEPs typically memorize the population of all executed policies and their outcomes, this enables reuse of experimentations across multiple goals.
- A *Meta-Policy* $\Pi : \mathcal{T}, \mathcal{C} \mapsto \Theta$ which is a mechanism to approximately solve the minimization problem $\Pi(\tau, c) = \arg\min_\theta C_\tau(\tilde{D}(\theta, c))$, where $\tilde{D}$ is a running forward model (approximating $D$), trained on-line during exploration.

In some applications, a *de-facto* ensemble of such tools can be used. For example, in the case where $\mathcal{O}$ is an Euclidean space, we can allow the agent to set goals in the Outcome Space $\mathcal{T} = \mathcal{O}$, in which case for every goal $\tau$ we can consider a Goal-parameterized cost function $C_\tau(o) = \|\tau - o\|$ where $\|.\|$ is a similarity metric. In the case of the Arm-Ball problem, the final position of the ball can be used as Outcome Space, hence the Euclidean distance between the goal position and the final ball position at the end of the episode can be used as Goal-parameterized cost function (but one could equally choose the full trajectories of the ball as outcomes and goals, and an associated similarity metric).

---

[9]See Section 3 for details.

Algorithmic architecture 2 describes the main steps of Intrinsically Motivated Goal Exploration Processes using these tools[10]:

**Bootstrapping phase:** Sampling a few policy parameters (called Random Parametrization Exploration, RPE), observing the starting context and the resulting outcome, to initialize a memory of experiments ($\mathcal{H} = \{(c_i, \theta_i, o_i)\}$) and a regressor $\tilde{D}_{running}$ approximating the phenomenon dynamics.

**Goal exploration phase:** Stochastically mixing random policy exploration with goal exploration. In goal exploration, one first observes the context $c$ and then samples a goal $\tau$ using goal policy $\gamma$ (this goal policy can be a random stationary distribution, as in experiments below, or a contextual multi-armed bandit maximizing information gain or competence progress, see (Baranes & Oudeyer, 2013)). Then, a meta-policy algorithm $\Pi$ is used to search the parameterization $\theta$ minimizing the Goal-parameterized cost function $C_\tau$, i.e. it computes $\theta = \arg\min_\theta C_\tau(\tilde{D}_{running}(\theta, c))$. This process is typically initialized by searching the parameter $\theta_{init}$ in $\mathcal{H}$ such that the corresponding $c_{init}$ is in the neighborhood of $c$ and $C_\tau(o_{init})$ is minimized. Then, this initial guess is improved using an optimization algorithm (e.g. L-BFGS) over the regressor $\tilde{D}_{running}$. The resulting policy $\theta$ is executed, and the outcome $o$ is observed. The observation $(c, \theta, o)$ is then used to update $\mathcal{H}$ and $\tilde{D}_{running}$.

This procedure has been experimentally shown to enable sample efficient exploration in high-dimensional continuous action robotic setups, enabling in turn to learn repertoires of skills in complex physical setups with object manipulations using tools (Forestier & Oudeyer, 2016; Forestier et al., 2017) or soft deformable objects (Nguyen & Oudeyer, 2014).

Nevertheless, two issues arise when it comes to using these algorithms in real-life setups, and within a fully autonomous learning approach. First, there are many real world cases where providing an Outcome Space (in which to make observations and sample goals, so this is also the Goal Space) to the agent is difficult, since the designer may not himself understand well the space that the robot is learning about. The approach taken until now (Forestier et al., 2017), was to create an external program which extracted information out of images, such as tracking all objects positions. This information was presented to the agent as a point in $[0, 1]^n$, which was hence considered as an Outcome Space. In such complex environments, the designer may not know what is actually feasible or not for the robot, and the Outcome space may contain many unfeasible goals. This is the reason why advanced mechanisms for sampling goals and discovering which ones are actually feasible have been designed (Baranes & Oudeyer, 2013; Forestier et al., 2017). Second, a system where the engineer designs the representation of an Outcome Space space is limited in its autonomy. A question arising from this is: can we design a mechanism that allows the agent to construct an Outcome Space that leads to efficient exploration by the mean of examples? Representation Learning methods, in particular Deep Learning algorithms, constitute a natural approach to this problem as it has shown outstanding performances in learning representations for images. In the next two sections, we present an update of the IMGEP architecture that includes a goal space representation learning stage, as well as various Deep Representation Learning algorithms tested: Autoencoders along with their more recent Variational counter-parts.

## 2.2 Unsupervised Goal Representation Learning for IMGEP

In order to enable goal space representation learning within the IMGEP framework, we propose to add a first stage of unsupervised perceptual learning (called UGL) before the goal exploration stage, leading to the new IMGEP-UGL architecture described in Algorithmic Architecture 1. In the passive perceptual learning stage (UGL, lines 2-8), the learner passively observes the unknown phenomenon by collecting samples $x_i$ of raw sensor values as the world changes. The architecture is neutral with regards to how these world changes are produced, but as argued in the introduction, one can see them as coming from actions of other agents in the environment. Then, this database of $x_i$ observations is used to train an unsupervised learning algorithm (e.g. VAE, Isomap) to learn an embedding function $\tilde{\mathcal{R}}$ which maps the high-dimensional raw sensor observations onto a lower-

---

[10]IMGEPs characterize an architecture and not an algorithm as several of the steps of this architecture can be implemented in multiple ways, for e.g. depending on which regression or meta-policy algorithms are implemented

dimensional representation $o$. Also, a kernel density estimator $KDE$ estimates the distribution $p_{kde}(o)$ of observed world changes projected in the embedding. Then, in the goal exploration stage (lines 9-26), this lower-dimensional representation $o$ is used as the outcome and goal space, and the distribution $p_{kde}(o)$ is used as a stochastic goal policy, within a standard IMGEP process (see above).

---

**Algorithmic Architecture 1:** Intrinsically Motivated Goal Exploration Process with Unsupervised Goal Representation Learning (IMGEP-UGL)

---

**Input:**

Regressor $\tilde{D}_{running}$, Goal Policy $\gamma$, Parameterized cost function $C_\tau$, Meta-Policy algorithm $\Pi$, Unsupervised representation learning algorithm $\mathcal{A}$ (e.g. AE, VAE, Isomap), Kernel Density Estimator algorithm $KDE$,
History $\mathcal{H}$, Random exploration ratio $\Gamma_e$

1  **begin**
2     **Passive perceptual learning stage (UGL):**
3     **for** *A fixed number of Observation iterations $n_r$* **do**
4         Observe the phenomenon with raw sensors to gather a sample $x_i$
5         Add this sample to a sample database $\mathcal{D} = \{x_i\}_{i \in [0, n_r]}$
6     Learn an embedding function $\tilde{R} : \ x \rightarrow o$ using algorithm $\mathcal{A}$ on data $\mathcal{D}$
7     Set $\mathcal{O} = \mathcal{T} = \tilde{R}(x)$
8     Estimate the outcome distribution $p_{kde}(o)$ from $\{\tilde{R}(x_i)\}_{i \in [0, 10000]}$ using algorithm $KDE$
9     Set the Goal Policy $\gamma = p_{kde}$ to be the estimated outcome distribution
10    **Goal exploration stage (IMGEP):**
11    **for** *A fixed number of Bootstrapping iterations* **do**
12        Observe context $c$
13        Sample $\theta \sim \mathcal{U}(\theta)$
14        Perform experiment and retrieve outcome from raw sensor signal $o = \tilde{R}(x)$
15        Update Regressor $\tilde{D}_{running}$ with tuple $\{c, \theta, o\}$
16        $\mathcal{H} = \mathcal{H} \cup \{c, \theta, o\}$
17    **for** *A fixed number of Exploration iterations* **do**
18        **if** $u \sim \mathcal{U}(0,1) < \Gamma_e$ **then**
19            Sample a random parameterization $\theta_i \sim p(\theta)$
20        **else**
21           Observe context $c$
22           Sample a goal $\tau \sim \gamma$
23           Compute $\theta = \arg \min_\theta C_\tau(\tilde{D}_{running}(\theta, c))$ using $\Pi$, $\tilde{D}_{running}$ and $\mathcal{H}$
24        Perform experiment and retrieve outcome from raw sensor signal $o = \tilde{R}(x)$
25        Update Regressor $\tilde{D}_{running}$ with a tuple $\{c, \theta, o\}$
26        Update Goal Policy $\gamma$, according to Intrinsic Motivation strategy
27        $\mathcal{H} = \mathcal{H} \cup \{c, \theta, o\}$
28  **return** *The forward model $\tilde{D}_{running}$, the history $\mathcal{H}$ and the embedding $\tilde{R}$*

---

## 2.3   Representation Learning Algorithms and Density Estimation for the UGL Stage

As IMGEP-UGL is an algorithmic architecture, it can be implemented with several algorithmic variants depending on which unsupervised learning algorithm is used in the UGL phase. We experimented over different deep and classical Representation Learning algorithms for the UGL phase. We rapidly outline these algorithms here. For a more in-depth introduction to those models, the reader can refer to Appendix B which contains details on the derivations of the different Cost Functions and Architectures of the Deep Neural Networks based models.

**Auto-Encoders (AEs)**   are a particular type of Feed-Forward Neural Networks that were introduced in the early hours of neural networks (Bourlard & Kamp, 1988). They are trained to output a reconstruction $\tilde{\mathbf{x}}$ of the input vector $\mathbf{x}$ of dimension $D$, through a representation layer of size $d < D$. They can be trained in an unsupervised manner using a large dataset of unlabeled samples $\mathcal{D} = \{\mathbf{x}^{(i)}\}_{i \in \{0...N\}}$. Their main interest lies in their ability to model the statistical regularities existing in the data. Indeed, during training, the network learns the regularities allowing to encode most of the information existing in the input in a more compact representation. Put differently, AEs can be seen as learning a non-linear compression for data coming from an unknown distribution. Those models can be trained using different algorithms, the most simple being Stochastic Gradient Descent (SGD), to minimize a loss function $\mathcal{J}(\mathcal{D})$ that penalizes differences between $\tilde{\mathbf{x}}$ and $\mathbf{x}$ for all samples in $\mathcal{D}$.

**Variational Auto-Encoders (VAEs)**   are a recent alternative to classic AEs (Rezende et al., 2014; Kingma & Ba, 2015), that can be seen as an extension to a stochastic encoding. The argument underlying this model is slightly more involved than the simple approach taken for AEs, and relies on a statistical standpoint presented in Appendix B. In practice, this model simplifies to an architecture very similar to an AE, differing only in the fact that the encoder $f_\theta$ outputs the parameters $\mu$ and $\sigma$ of a multivariate Gaussian distribution $\mathcal{N}(\mu, diag(\sigma^2))$ with diagonal covariance matrix, from which the representation $\mathbf{z}$ is sampled. Moreover, an extra term is added to the Cost Function, to condition the distribution of $\mathbf{z}$ in the representation space. Under the restriction that a factorial Gaussian is used, the neural network can be made fully differentiable thanks to a *reparameterization trick*, making it possible to use SGD for training.

In practice VAEs tend to yield smooth representations of the data, and are faster to converge than AEs from our experiments. Despite these interesting properties, the derivation of the actual cost function relies mostly on the assumption that the factors can be described by a factorial Gaussian distribution. This hypothesis can be largely erroneous, for example if one of the factors is periodic, multi-modal, or discrete. In practice our experiments showed that even if training could converge for non-Gaussian factors, it tends to be slower and to yield poorly conditioned representations.

**Normalizing Flow**   proposes a way to overcome this restriction on distribution, by allowing more expressive ones (Rezende & Mohamed, 2015). It uses the classic rule of change of variables for random variables, which states that considering a random variable $\mathbf{z}_0 \sim q(\mathbf{z}_0)$, and an invertible transformation $t : \mathbb{R}^d \mapsto \mathbb{R}^d$, if $\mathbf{z} = t(\mathbf{z}_0)$ then $q(\mathbf{z}) = q(\mathbf{z}_0)|\det \partial t/\partial \mathbf{z}_0|^{-1}$. Using this, we can chain multiple transformations $t_1, t_2, \ldots, t_K$ to produce a new random variable $\mathbf{z}_K = t_K \circ \cdots \circ t_2 \circ t_1(\mathbf{z}_0)$. One particularly interesting transformation is the *Radial Flow*, which allows to radially contract and expand a distribution as can be seen in Figure 5 in Appendix. This transformation seems to give the required flexibility to encode periodic factors.

**Isomap**   is a classical approach of Multi-Dimensional Scaling (Kruskal, 1964) a procedure allowing to embed a set of $N$-dimensional points in a $n$ dimensional space, with $N > n$, minimizing the *Kruskal Stress*, which measures the distortion induced by the embedding in the pairwise Euclidean distances. This algorithm results in an embedding whose pairwise distances are roughly the same as in the initial space. Isomap (Tenenbaum et al., 2000) goes further by assuming that the data lies in the vicinity of a lower dimensional manifold. Hence, it replaces the pairwise Euclidean distances in the input space by an approximate pairwise geodesic distance, computed by the Dijkstra's Shortest Path algorithm on a $\kappa$ nearest-neighbors graph.

**Principal Component Analysis**   is an ubiquitous procedure (Pearson, 1901) which, for a set of data points, allows to find the orthogonal transformation that yields linearly uncorrelated data. This transformation is found by taking the principal axis of the covariance matrix of the data, leading to a representation whose variance is in decreasing order along dimensions. This procedure can be used to reduce dimensionality, by taking only the first $n$ dimensions of the transformed data.

**Estimation of sampling distribution:**   Since the Outcome Space $\mathcal{O}$ was learned by the agent, it had no prior knowledge of $p(o)$ for $o \in \mathcal{O}$. We used a *Gaussian Kernel Density Estimation* (KDE) (Parzen, 1962; Rosenblatt, 1956) to estimate this distribution from the projection of the images observed by the agent, into the learned goal space representation. Kernel Density Estimation allows

to estimate the continuous density function (cdf) $f(o)$ out of a discrete set of samples $\{o_i\}_{i\in\{1,...,n\}}$ drown from distribution $p(o)$. The estimated cdf is computed using the following equation:

$$\hat{f}_{\mathbf{H}}(o) = \frac{1}{n}\sum_{i=1}^{n} K_{\mathbf{H}}(o - o_i), \tag{1}$$

with $K(\cdot)$ a kernel function and $\mathbf{H}$ a bandwidth $d \times d$ matrix (d the dimension of $\mathcal{O}$). In our case, we used a Gaussian Kernel:

$$K_{\mathbf{H}}(o) = (2\pi)^{-\frac{d}{2}}|\mathbf{H}|^{-\frac{1}{2}}e^{-\frac{1}{2}o^T\mathbf{H}^{-1}o}, \tag{2}$$

with the bandwidth matrix $\mathbf{H}$ equaling the covariance matrix of the set of points, rescaled by factor $n^{-\frac{1}{d+4}}$, with $n$ the number of samples, as proposed in Scott (1992).

## 3 EXPERIMENTS

We conducted experiments to address the following questions in the context of two simulated environments:

- Is it possible for an IMGEP-UGL implementation to produce a Goal Space representation yielding an exploration dynamics as efficient as the dynamics produced by an IMGEP implementation using engineered goal space representations? Here, the dynamics of exploration is measured through the *KL Coverage* defined thereafter.
- What is the impact of the target embedding dimensionality provided to these algorithms?
- Are there differences in exploration dynamics when one uses different unsupervised learning algorithms (Isomap-KDE, PCA-KDE, AE-KDE, VAE-KDE, VAE-GP, RFVAE-GP, RFVAE-KDE) as various UGL component of IMGEP-UGL?

We now present in depth the experimental campaign we performed[11].

**Environments:** We experimented on two different **Simulated Environments** derived from the Arm-Ball benchmark represented in Figure 1, namely the *Arm-Ball* and the *Arm-Arrow* environments, in which a 7-joint arm, controlled by a 21 continuous dimension Dynamic Movement Primitives (DMP) (Ijspeert et al., 2013) controller, evolves in an environment containing an object it can handle and move around in the scene. In the case of IMGEP-UGL learners, the scene is perceived as a 70x70 pixel image. For the UGL phase, we used the following mechanism to generate the distribution of samples $x_i$: the object was moved randomly uniformly over $[-1, 1]^2$ for ArmBall, and over $[-1, 1]^2 \times [0, 2\pi]$ for ArmArrow, and the corresponding images were generated and provided as an observable sample to IMGEP-UGL learners. Note that the physically reachable space (i.e. the largest space the arm can move the object to) is the disk centered on 0 and of radius 1: this means that the distribution of object movements observed by the learner is slightly larger than the actual space of moves that learners can produce themselves (and learners have no knowledge of which subspace corresponds to physically feasible outcomes). The environments are presented in depth in Appendix C.

**Algorithmic Instantiation of the IMGEP-UGL Architecture:** We experimented over the following Representation Learning Algorithms for the UGL component: *Auto-Encoders* with $KDE$ (RGE-AE), *Variational Auto-Encoders* with $KDE$ (RGE-VAE), *Variational Auto-Encoders* using the associated Gaussian prior for sampling goal instead of $KDE$ (RGE-VAE-GP), *Radial Flow Variational Auto-Encoders* with $KDE$ (RGE-RFVAE), *Radial Flow Variational Auto-Encoders* using the associated Gaussian prior for sampling goal (RGE-RFVAE-GP), *Isomap* (RGE-Isomap) (Tenenbaum et al., 2000) and *Principal Component Analysis* (RGE-Isomap).

Regarding the classical IMGEP components, we considered the following elements:

- **Context Space** $\mathcal{C} = \varnothing$**:** In the implemented environments, the initial positions of the arm and the object were reset at each episode[12]. Consequently, the context was not observed nor accounted for by the agent.

---

[11]The code to reproduce the experiments is available at
https://github.com/flowersteam/Unsupervised_Goal_Space_Learning
[12]This makes the experiment faster but does not affect the conclusion of the results.

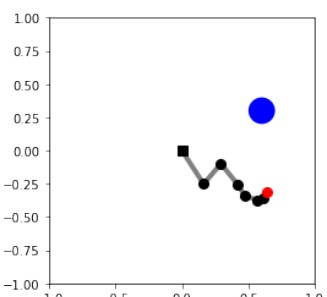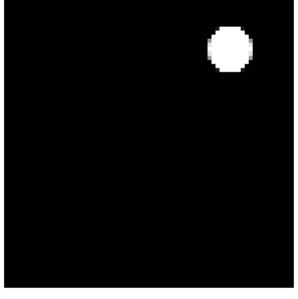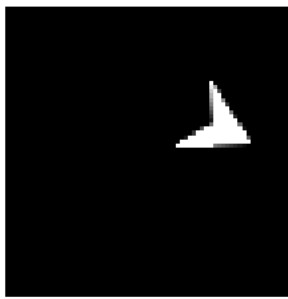

Figure 1: Left: The Arm-Ball environment with a 7 DOF arm, controlled by a 21D continuous actions DMP controller, that can stick and move the ball if the arm tip touches it (on the left). Right: rendered 70x70 images used as raw signals representing the end position of the objects for Arm-Ball (on the center) and Arm-Arrow (on the right) environments. The arm is not visible to learners.

- **Parameterization Space** $\Theta = [0, 1]^{21}$**:** During the experiments, we used DMP controllers as parameterized policies to generate time-bounded motor actions sequences. Since the DMP controller was parameterized by 3 basis functions for each joint of the arm (7), the parameterization of the controller was represented by a point in $[0, 1]^{3 \times 7}$.
- **Outcome Space** $\mathcal{O} \subset \mathbb{R}^l$**:** The Outcome Space is the subspace of $\mathbb{R}^l$ spanned by the embedding representations of the ensemble of images observed in the first phase of learning. For the RGE-EFR algorithm, $l = 2$ in ArmBall and $l = 3$ in ArmArrow. For IMGEP-UGL algorithms, as the representation learning algorithms used in the UGL stage require a parameter specifying the maximum dimensionality of the target embedding, we considered two cases in experiments: 1) $l = 10$, which is 5 times larger than the true manifold dimension for ArmBall, and 3.3 times larger for ArmArrow (the algorithm is not supposed to know this, so testing the performance with larger embedding dimension is key); 2) $l = 2$ for ArmBall, and $l = 3$ for ArmArrow, which is the same dimensionality as the true dimensions of these manifolds.
- **Goal Space** $\mathcal{T} = \mathcal{O}$ **:** The Goal Space was taken to equate the Outcome Space.
- **Goal-Parameterized Cost function** $C_\tau(\cdot) = \|\tau - \cdot\|_2$ **:** Sampling goals in the Outcome Space allows us to use the Euclidean distance as Goal-parameterized cost function.

Considering those elements, we used the instantiation of the IMGEP architecture represented in Appendix D in Algorithm 3. We implemented a goal sampling strategy known as *Random Goal Exploration* (RGE), which consists, given a stationary distribution over the Outcome Space $p(o)$, in sampling a random goal $o \sim p(o)$ each time (note that this stationary distribution $p(o)$ is learnt in the UGL stage for IMGEP-UGL implementations). We used a simple $k$-neighbors regressor to implement the running forward model $\tilde{D}$, and the Meta-Policy mechanism consisted in returning the nearest achieved outcome in the outcome space, and taking the same parameterization perturbed by an exploration noise (which has proved to be a very strong baseline in IMGEP architectures in previous works (Baranes & Oudeyer, 2013; Forestier & Oudeyer, 2016)).

**Exploration Performance Measure:** In this article, the central property we are interested in is the dynamics and quality of exploration of the outcome space, characterizing the evolution of the distribution of discovered outcomes, i.e. the diversity of effects that the learner discovers how to produce. In order to characterize this exploration dynamics quantitatively, we monitored a measure which we refer to as *Kullback-Leibler Coverage* (KLC). At a given point in time during exploration, this measure computes the KL-divergence between the distribution of the outcomes produced so far, with a uniform distribution of outcomes in the space of physically possible outcomes (which is known by the experimenter, but unknown by the learner). To compute it, we use a normalized histogram of the explored outcomes, with 30 bins per dimension, which we refer to as $E$, and we compute its Kullback Leibler Divergence with the normalized histogram of attainable points which

we refer to as $A$:

$$KLC = \mathbb{D}_{KL}[E\|A] = \sum_{i=1}^{30} E(i) \log \frac{E(i)}{A(i)}.$$

We emphasize that, when computed against a uniform distribution, the KLC measure is a proxy for the (opposite) Entropy of the $E$ distribution. Nevertheless, we prefer to keep it under the divergence form, as the $A$ distribution allows to define what the experimenter considers to be a good exploration distribution. In the case of this study, we consider a uniform distribution of explored locations over the attainable domain, to be the best exploration distribution achievable.

**Baseline algorithms:** We are using two natural baseline algorithms for evaluating the exploration dynamics of our IMGEP-UGL algorithmic implementations :

- **Random Goal Exploration with Engineered Features Representations (RGE-EFR):** This is an IMGEP implementation using a goal/outcome space with handcrafted features that directly encode the underlying structure of environments: for Arm-Ball, this is the 2D position of the ball in $[0, 1]^2$, and for Arm-Arrow this is the 2D position and the 1D orientation of the arrow in $[0, 1]^3$. This algorithm is also given the prior knowledge of $p(o) = \mathcal{U}(\mathcal{O})$. All other aspects of the IMGEP (regressor, meta-policy, other parameters) are identical to IMGEP-UGL implementations. This algorithm is known to provide highly efficient exploration dynamics in these environments (Forestier & Oudeyer, 2016).
- **Random Parameterization Exploration (RPE):** The Random Parameterization Exploration approach does not use an Outcome Space, nor a Goal Policy, and only samples a random parameterization $\theta \sim \mathcal{U}(\Theta)$ at each episode. We expected this algorithm to lower bound the performances of our novel architecture.

## 4 RESULTS

We first study the exploration dynamics of all IMGEP-UGL algorithms, comparing them to the baselines and among themselves. Then, we study specifically the impact of the target embedding dimension (latent space) for the UGL implementations, by observing what exploration dynamics is produced in two cases:

- Using a target dimension larger than the true dimension ($l = 10$)
- Providing the true embedding dimension to the UGL implementations ($l = 2, 3$)

Finally, we specifically study RGE-VAE, using the intrinsic Gaussian prior of these algorithms to replace the $KDE$ estimator of $p(O)$ in the UGL part.

**Exploration Performances:** In Figure 2, we can see the evolution of the KLC through exploration epochs (one exploration epoch is defined as one experimentation/roll-out of a parameter $\theta$). We can see that for both environments, and all values of latent spaces, all IMGEP-UGL algorithms, except RGE-RFVAE, achieve similar or better performance (both in terms of asymptotic KLC and speed to reach it) than the RGE-EFR algorithm using engineered Goal Space features, and much better performance than the RPE algorithm.

Figure 3 (see also Figure 8 and 9 in Appendix) show details of the evolution of discovered outcomes in ArmBall (final ball positions after the end of a policy roll-out) and corresponding KLC measures for individual runs with various algorithms. It also shows the evolution of the number of times learners managed to move the ball, which is considered in the KLC measure but not easily visible in the displayed set of outcomes in Figure 3. For instance, we observe that both RPE (Figure 3(a)) and RGE-RFVAE (Figure 3(c)) algorithms perform poorly: they discover very few policies moving the ball at all (pink curves), and these discovered ball moves cover only a small part of the physically possible outcome space. On the contrary, both RGE-EFR (handcrafted features) and RGE-VAE (learned goal space representation with VAE) perform very well, and the KLC of RGE-VAE is even better than the KLC of RGE-EFR, due to the fact that RGE-VAE has discovered more policies (around 2400) that move the ball than RGE-EFR (around 1600, pink curve).

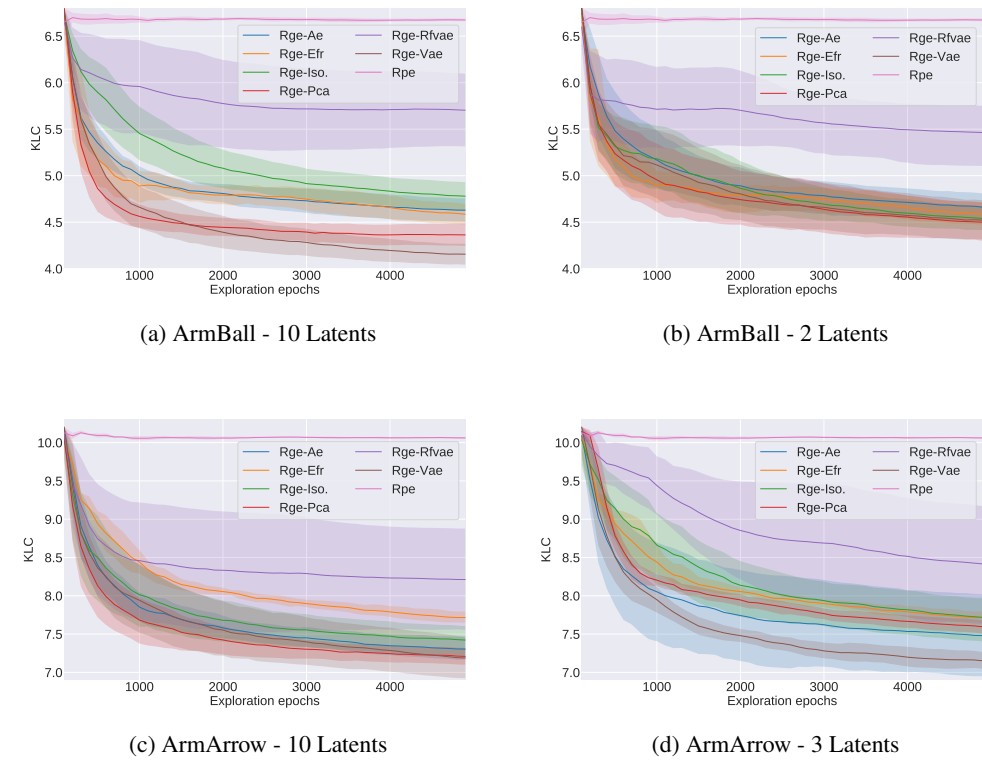

Figure 2: KL Coverage through epochs for different algorithms on ArmBall and ArmArrow environments. The exploration performance was assessed for both an over-complete representation (10 latent dimensions), and a complete representation (2 and 3 latent dimensions). The shaded area represent a 90% confidence interval estimated from 5 run of the different algorithms.

**Impact of target latent space size in IMGEP-UGL algorithms** On the ArmBall problem, we observe that if one provides the true target embedding dimension ($l = 2$) to IMGEP-UGL implementations, RGE-Isomap is slightly improving (getting quasi-identical to RGE-EFR), RGE-AE does not change (remains quasi-identical to RGE-EFR), but the performance of RGE-PCA and RGE-VAE is degraded. For ArmArrow, the effect is similar: IMGEP-UGL algorithms with a larger target embedding dimension ($l = 10$) than the true dimensionality all perform better than RGE-EFR (except RGE-RFVAE which is worse in all cases), while when $l = 2$ only RGE-VAE is significantly better than RGE-EFR. In Appendix F, more examples of exploration curves with attached exploration scatters are shown. For most example runs, increasing the target embedding dimension enables learners to discover more policies moving the ball and, in these cases, the discovered outcomes are more concentrated towards the external boundary of the discus of physically possible outcomes. This behavior, where increasing the target embedding dimension improves the KLC while biasing the discovered outcome towards the boundary the feasible goals, can be understood as a consequence of the following well-known general property of IMGEPs: if goals are sampled outside the convex hull of outcomes already discovered, this has the side-effect of biasing exploration towards policies that will produce outcomes beyond this convex hull (until the boundary of feasible outcomes is reached). Here, as observations in the UGL phase were generated by uniformly moving the objects on the square $[-1, 1]^2$, while the feasible outcome space was the smaller discus of radius 1, goal sampling happened in a distribution of outcomes larger than the feasible outcome space. As one increases the embedding space dimensionality, the ratio between the volume of the corresponding hyper-cube and hyper-discus increases, in turn increasing the probability to sample goals outside the feasible space, which has the side effect of fostering the discovery of novel outcomes and biasing exploration towards the boundaries.

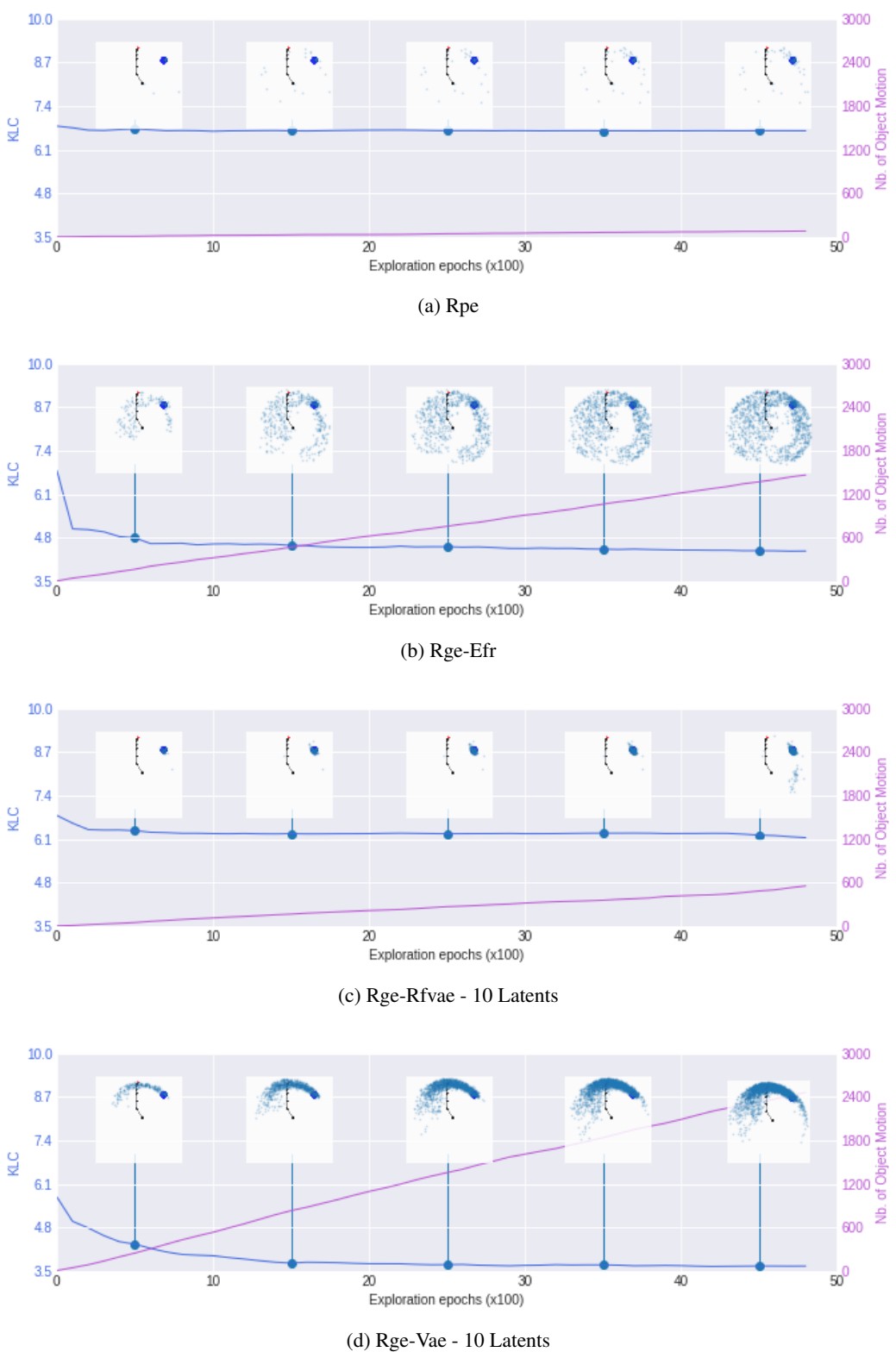

(a) Rpe

(b) Rge-Efr

(c) Rge-Rfvae - 10 Latents

(d) Rge-Vae - 10 Latents

Figure 3: Examples of achieved outcomes related with the evolution of KL-Coverage in the ArmBall environments. The number of times the ball was effectively handled is also represented.

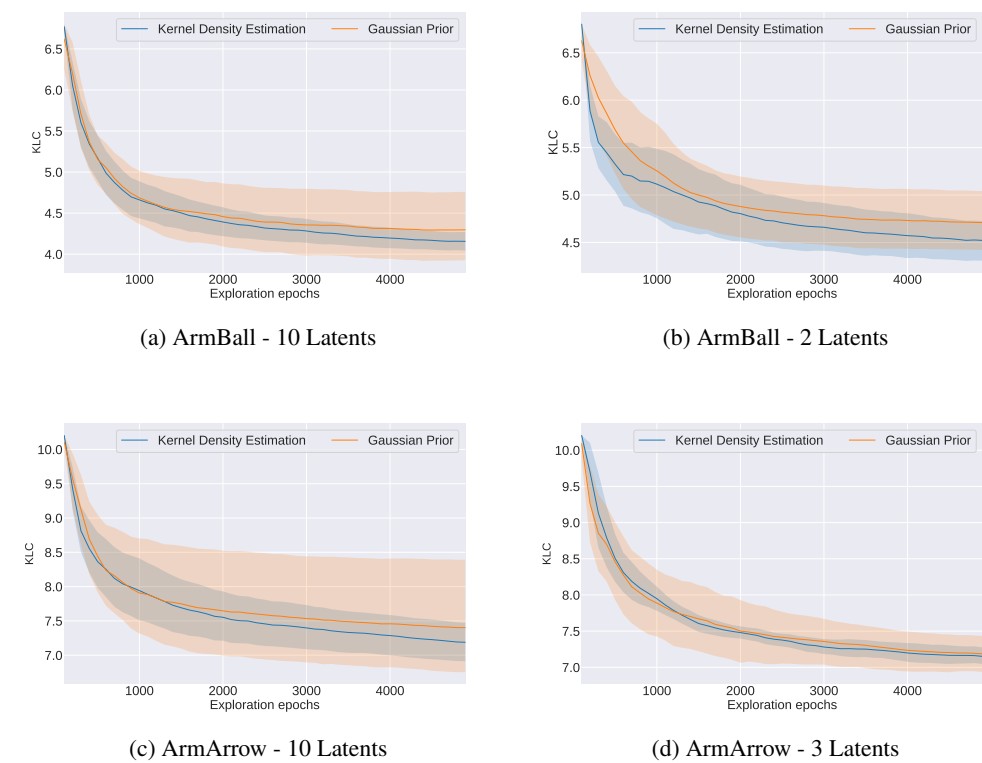

Figure 4: Evolution of the Exploration Ratio for RGE-VAE using KDE or Isotropic Gaussian prior. The curves show the mean and standard deviation over 5 independent runs of each condition.

**Impact of Sampling Kernel Density Estimation**    Another factor impacting the exploration assessed during our experiments was the importance of the distribution used as stationary Goal Policy. If, in most cases, the representation algorithm gives no particular prior knowledge of $p(o)$, in the case of Variational Auto-Encoders, it is assumed in the derivation that $p(o) = \mathcal{N}(0, I)$. Hence, the isotropic Gaussian distribution is a better candidate stationary Goal Policy than Kernel Density Estimation. Figure 4 shows a comparison between exploration performances achieved with RGE-VAE using a KDE distribution or an isotropic Gaussian as Goal Policy. The performance is not significantly different from the isotropic Gaussian case. Our experiments showed that convergence on the KL term of the loss can be more or less quick depending on the initialization. Since we used a number of iterations as stopping criterion for training (based on early experiments), we found that sometimes, at stop, the divergence was still pretty high despite achieving a low reconstruction error. In those cases the representation was not be perfectly matching an isotropic Gaussian, which could lead to a goal sampling bias when using the isotropic Gaussian Goal Policy.

## 5    CONCLUSION

In this paper, we proposed a new Intrinsically Motivated Goal Exploration architecture with Unsupervised Learning of Goal spaces (IMGEP-UGL). Here, the Outcome Space (also used as Goal Space) representation is learned using passive observations of world changes through low-level raw sensors (e.g. movements of objects caused by another agent and perceived at the pixel level). Within the perspective of research on Intrinsically Motivated Goal Exploration started a decade ago (Oudeyer & Kaplan, 2007; Baranes & Oudeyer, 2013), and considering the fundamental problem of how AI agents can autonomously explore environments and skills by setting their own goals, this new architecture constitutes a milestone as it is to our knowledge the first goal exploration architecture where the goal space representation is learned, as opposed to hand-crafted.

Furthermore, we have shown in two simulated environments (involving a high-dimensional continuous action arm) that this new architecture can be successfully implemented using multiple kinds of unsupervised learning algorithms, including recent advanced deep neural network algorithms like Variational Auto-Encoders. This flexibility opens the possibility to benefit from future advances in unsupervised representation learning research. Yet, our experiments have shown that all algorithms we tried (except RGE-RFVAE) can compete with an IMGEP implementation using engineered feature representations. We also showed, in the context of our test environments, that providing to IMGEP-UGL algorithms a target embedding dimension larger than the true dimensionality of the phenomenon can be beneficial through leveraging exploration dynamics properties of IMGEPs. Though we must investigate more systematically the extent of this effect, this is encouraging from an autonomous learning perspective, as one should not assume that the learner initially knows the target dimensionality.

**Limits and future work.** The experiments presented here were limited to a fairly restricted set of environments. Experimenting over a larger set of environments would improve our understanding of IMGEP-UGL algorithms in general. In particular, a potential challenge is to consider environments where multiple objects/entities can be independently controlled, or where some objects/entities are not controllable (e.g. animate entities). In these cases, previous work on IMGEPs has shown that random Goal Policies should be either replaced by modular Goal Policies (considering a modular goal space representation, see Forestier et al. (2017)), or by active Goal Policies which adaptively focus the sampling of goals in subregions of the Goal Space where the competence progress is maximal (Baranes & Oudeyer, 2013). For learning modular representations of Goal Spaces, an interesting avenue of investigations could be the use of the Independently Controllable Factors approach proposed in (Thomas et al., 2017).

Finally, in this paper, we only studied a learning scenario where representation learning happens first in a passive perceptual learning stage, and is then fixed during a second stage of autonomous goal exploration. While this was here motivated both by analogies to infant development and to facilitate evaluation, the ability to incrementally and jointly learn an outcome space representation and explore the world is a stimulating topic for future work.

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

# Appendix

## A  INTRINSICALLY MOTIVATED GOAL EXPLORATION PROCESS

Intrinsically Motivated Goal Exploration Processes are algorithmic architectures that can be instantiated into different exploration algorithms depending on the problem to explore. The general architecture is represented in Algorithm 2.

---

**Algorithmic Architecture 2:** Intrinsically Motivated Goal Exploration Strategy

---

**Input:**

Regressor $\tilde{D}_{running}$, Goal Policy $\gamma$, Meta-Policy algorithm $\Pi$, History $\mathcal{H}$, Random exploration ratio $\Gamma_e$

1 **begin**
2    **for** *A fixed number of Bootstrapping iterations* **do**
3       Observe context $c$
4       Sample $\theta \sim \mathcal{U}(\theta)$
5       Perform experiment and retrieve outcome $o$
6       Update Regressor $\tilde{D}_{running}$ with tuple $\{c, \theta, o\}$
7       $\mathcal{H} = \mathcal{H} \cup \{c, \theta, o\}$
8    **for** *A fixed number of Exploration iterations* **do**
9       **if** $u \sim \mathcal{U}(0,1) < \Gamma_e$ **then**
10          Sample a random parameterization $\theta_i \sim p(\theta)$
11       **else**
12          Observe context $c$
13          Sample a goal $\tau \sim \gamma$
14          Compute $\theta = \arg\min_\theta C_\tau(\tilde{D}_{running}(\theta, c))$ using $\Pi$, $\tilde{D}_{running}$ and $\mathcal{H}$
15       Perform experiment and retrieve outcome $o$
16       Update Regressor $\tilde{D}_{running}$ with the tuple $\{c, \theta, o\}$
17       Update Goal Policy $\gamma$ according to Intrinsic Motivation strategy
18       $\mathcal{H} = \mathcal{H} \cup \{c, \theta, o\}$
19 **return** *The forward model $\tilde{D}_{running}$ and the history $\mathcal{H}$*

---

## B  DEEP REPRESENTATION LEARNING ALGORITHMS

The cost functions used to train the different Deep Representation Learning algorithms used in this paper can be motivated by a few theoretical arguments summarized below.

**Auto-Encoders (AEs)**   The choice of the cost function can be motivated by considering the network as composed of:

- An *encoder* network parameterized by weights $\theta$ that maps an input $\mathbf{x}$ to its deterministic representation $\mathbf{z} = f_\theta(\mathbf{x})$.

- A *decoder* network parameterized by weights $\phi$ that maps a representation $\mathbf{z}$ to a vector $\xi$ parameterizing a distribution $p_\xi(\mathbf{x}|\mathbf{z})$ with $\xi = g_\phi(\mathbf{z})$.

Under this stochastic decoding assumption, the Maximum Likelihood principle is used to train the model, i.e. AEs can maximize the likelihood of data under the model. In the case of Auto-Encoders, this principle is compatible with gradient descent, and we can use the negative log-likelihood as a cost function to be minimized. If input $\mathbf{x}$ is binary valued, $p(\mathbf{x}|\mathbf{z})$ is assumed to follow a multivariate

Bernouilli distribution of $\xi$ parameters [13], and the log likelihood of the dataset $\mathcal{D}$ is expressed as:

$$\log \mathfrak{L}(\mathcal{D}) = \sum_{i=1}^{N} \log p(\mathbf{x}^{(i)}|\xi^{(i)}) = \sum_{i=1}^{N} \sum_{k=1}^{D} \left[ x_k^{(i)} \log \xi_k^{(i)} + (1 - x_k^{(i)}) \log(1 - \xi_k^{(i)}) \right], \quad (3)$$

with $\xi^{(i)} = g_\phi(f_\theta(\mathbf{x}^{(i)}))$. For a binary valued input vector $\mathbf{x}^{(i)}$, the unitary Cost Function to minimize is:

$$\mathcal{J}(\theta, \phi, \mathbf{x}^{(i)}) = -\sum_{d=1}^{D} \left[ x_d^{(i)} \log(g_\phi(f_\theta(\mathbf{x}^{(i)}))_d) + (1 - x_d^{(i)}) \log(1 - g_\phi(f_\theta(\mathbf{x}^{(i)}))_d) \right], \quad (4)$$

provided that $f_\theta$ is the encoder part of the architecture and $g_\phi$ is the decoding part of the architecture. This Cost Function can be minimized using Stochastic Gradient Descent (Bottou, 1998), or more advanced optimizers such as Adagrad (Duchi et al., 2011) or Adam (Kingma & Ba, 2015).

Depending on the depth of the network[14], those architectures can prove difficult to train using vanilla Stochastic Gradient Descent. A particularly successful procedure to overcome this difficulty is to greedily train each pairs of encoding-decoding layers and stacking those to sequentially form the complete network. This procedure, known as stacked AEs, accelerates convergence. But it has shown bad results with our problem, and thus was discarded for the sake of clarity.

**Variational Auto-Encoders (VAEs)** If we assume that the observed data are realizations of a random variable $\mathbf{x} \sim p(\mathbf{x}|\psi)$, we can hypothesize that they are conditioned by a random vector of independent factors $\mathbf{z} \sim p(\mathbf{z}|\psi)$. In this setting, learning the model would amount to searching the parameters $\psi$ of both distributions. We might use the same principle of maximum likelihood as before to find the best parameters by computing the likelihood $\log \mathfrak{L}(\mathcal{D}) = \sum_{i=1}^{N} \log p(\mathbf{x}^{(i)}|\psi)$ by using the fact that $p(\mathbf{x}|\psi) = \int p(\mathbf{x}, \mathbf{z}|\psi) d\mathbf{z} = \int p(\mathbf{x}|\mathbf{z}, \psi) p(\mathbf{z}|\psi) d\mathbf{z}$. Unfortunately, in most cases, this integral is intractable and cannot be approximated by Monte-Carlo sampling in reasonable time. To overcome this problem, we can introduce an arbitrary distribution $q(\mathbf{z}|\mathbf{x}, \chi)$ and remark that the following holds:

$$\log p(\mathbf{x}|\psi) = \mathcal{L}(q, \psi) + \mathbb{D}_{KL}[q(\mathbf{z}|\mathbf{x}, \chi)\|p(\mathbf{z}|\mathbf{x}, \psi), \quad (5)$$

with the Evidence Lower Bound being:

$$\mathcal{L}(q, \psi) = \underbrace{\mathbb{E}_{\mathbf{z} \sim q(\mathbf{z}|\mathbf{x}, \psi)}[\log p(\mathbf{x}|\mathbf{z}, \psi)]}_{a} - \underbrace{\mathbb{D}_{KL}[q(\mathbf{z}|\mathbf{x}, \chi)\|p(\mathbf{z}, \psi)]}_{b}. \quad (6)$$

Looking at Equation (5), we can see that since the KL divergence is non-negative, $\mathcal{L}(q, \psi) \leq \log p(\mathbf{x}|\psi) - \mathbb{D}_{KL}([q(\mathbf{z}|\mathbf{x}, \chi)\|p(\mathbf{z}|\mathbf{x}, \psi)]$ whatever the $q$ distribution, hence the name of Evidence Lower Bound (ELBO). Consequently, maximizing the ELBO have the effect to maximize the log likelihood, while minimizing the KL-Divergence between the approximate $q(\mathbf{z}|\mathbf{x})$ distribution, and the true unknown posterior $p(\mathbf{z}|\mathbf{x}, \psi)$. The approach taken by VAEs is to *learn* the parameters of both conditional distributions $p(\mathbf{x}|\mathbf{z}, \psi)$ and $q(\mathbf{z}|\mathbf{x}, \chi)$ as non-linear functions. Under some restricted conditions, Equation (6) can be turned into a valid cost function to train a neural network. First, we hypothesize that $q(\mathbf{z}|\mathbf{x}, \chi)$ and $p(\mathbf{z}|\psi)$ follow Multivariate Gaussian distributions with diagonal covariances, which allows us to compute the $b$ term in closed form. Second, using the Gaussian assumption on $q$, we can reparameterize the inner sampling operation by $\mathbf{z} = \mu + \sigma^2 \odot \epsilon$ with $\epsilon \sim \mathcal{N}(0, I)$. Using this trick, the Path-wise Derivative estimator can be used for the $a$ member of the ELBO. Under those conditions, and assuming that $p(\mathbf{x}|\psi)$ follows a Multivariate Bernouilli distribution, we can write the cost function used to train the neural network as:

$$\mathcal{J}(\psi, \chi, \mathbf{x}^{(i)}) = -\frac{1}{2} \sum_{j=1}^{J} (1 + \log(\sigma(\mathbf{x}^{(i)})_j^2) - \mu(\mathbf{x}^{(i)})_j^2 - \sigma(\mathbf{x}^{(i)})_j^2)$$

$$\quad (7)$$

$$- \sum_{k=1}^{D} \left[ x_k^{(i)} \log(g_\psi(f_\chi(\mathbf{x}^{(i)}))_k) + (1 - x_k^{(i)}) \log(1 - g_\psi(f_\chi(\mathbf{x}^{(i)}))_k) \right],$$

---

[13]This requires that the output layer uses a sigmoid function which restricts the values of output to $[0, 1]$.

[14]By depth here, we indicate the number of layers of the neural network.

where $f_\chi$ represents the encoding and sampling part of the architecture and $g_\psi$ represents the decoding part of the architecture. In essence, this derivation simplifies to the initial cost function used in AEs augmented by a term penalizing the divergence between $q(\mathbf{z}|\mathbf{x}, \chi)$ and the assumed prior that $p(\mathbf{x}|\psi) = \mathcal{N}(0, I)$.

**Normalizing Flow** overcomes the problem stated earlier, by permitting more expressive prior distributions (Rezende & Mohamed, 2015). It is based on the classic rule of change of variables for random variables. Considering a random variable $\mathbf{z}_0 \sim q(\mathbf{z}_0)$, and an invertible transformation $t : \mathbb{R}^d \mapsto \mathbb{R}^d$, if $\mathbf{z} = t(\mathbf{z}_0)$, then:

$$q(\mathbf{z}) = q(\mathbf{z}_0)\left| \det \frac{\partial t^{-1}}{\partial \mathbf{z}_0} \right| = q(\mathbf{z}_0)\left| \det \frac{\partial t}{\partial \mathbf{z}_0} \right|^{-1}. \tag{8}$$

We can then directly chain different invertible transformations $t_1, t_2, \ldots, t_K$ to produce a new random variable $\mathbf{z}_K = t_K \circ \cdots \circ t_2 \circ t_1(\mathbf{z}_0)$. In this case, we have:

$$\log q(\mathbf{z}_k) = \log\left( q(\mathbf{z}_0) \prod_{k=1}^{K} \left| \det \frac{\partial t_k}{\partial \mathbf{z}_{k-1}} \right|^{-1} \right) = \log q(\mathbf{z}_0) - \sum_{k=1}^{K} \log \left| \det \frac{\partial t_k}{\partial \mathbf{z}_{k-1}} \right|. \tag{9}$$

This formulation is interesting because the *Law Of The Unconscious Statistician* allows us to compute expectations over $q(\mathbf{z}_k)$ without having a precise knowledge of it:

$$\mathbb{E}_{\mathbf{z}_k \sim q(\mathbf{z}_k)}[h(\mathbf{z}_k)] = \mathbb{E}_{\mathbf{z}_0 \sim q(\mathbf{z}_0)}[h(t_k \circ \ldots t_2 \circ t_1(\mathbf{z}_0))], \tag{10}$$

provided that $h$ does not depends on $q(\mathbf{z}_k)$. Using this principle on the ELBO allows us to derive the following:

$$\begin{aligned}
\mathcal{L}(q, \theta, \phi) =& \mathbb{E}_{\mathbf{z}_0 \sim q(\mathbf{z}_0|\mathbf{x})}[\log p(\mathbf{x}|t_K \circ \ldots t_2 \circ t_1(\mathbf{z}_0))] \\
& -\mathbb{D}_{KL}[q(\mathbf{z}_0|\mathbf{x})\|p(\mathbf{z}_0)] \\
& +2\mathbb{E}_{\mathbf{z}_0 \sim q(\mathbf{z}_0|\mathbf{x})}\left[ \sum_{k=1}^{K} \log \left| \det \frac{\partial t_k}{\partial \mathbf{z}_{k-1}} \right| \right]
\end{aligned} \tag{11}$$

This is nothing more than the regular ELBO with an additional term concerning the log-determinant of the transformations. In practice, as before, we use $p(\mathbf{z}_0) = \mathcal{N}(\mathbf{z}_0; \mathbf{0}, \mathbf{I})$, and $q(\mathbf{z}_0|x) = \mathcal{N}(\mathbf{z}_0; \mu(x), diag(\sigma(x)^2))$. We only have to find out parameterized transformations $t$, whose parameters can be learned and have a defined log-determinant. Using *radial flow*, which is expressed as:

$$t(\mathbf{z}) = \mathbf{z} + \beta h(\alpha, \mathbf{r})(\mathbf{z} - \mathbf{c}), \tag{12}$$

where $\mathbf{r} = |\mathbf{z} - \mathbf{c}|$, $h(\alpha, \mathbf{r}) = \frac{1}{\alpha+\mathbf{r}}$ and $\alpha, \beta, \mathbf{c}$ are learnable parameters of the transformation, our cost function can be written as:

$$\begin{aligned}
\mathcal{J}(\psi, \chi, \mathbf{x}^{(i)}) =& -\frac{1}{2} \sum_{j=1}^{J}(1 + \log(\sigma(\mathbf{x}^{(i)})_j^2) - \mu(\mathbf{x}^{(i)})_j^2 - \sigma(\mathbf{x}^{(i)})_j^2) \\
& - \sum_{d=1}^{D} \left[ x_d^{(i)} \log(g_\psi(f_\chi(\mathbf{x}^{(i)}))_d) + (1 - x_d^{(i)}) \log(1 - g_\psi(f_\chi(\mathbf{x}^{(i)}))_d) \right] \\
& - 2 \sum_{k=1}^{K} \log[1 + \beta_k h(\alpha_k, \mathbf{r}))]^{D-1}[1 + \beta_k h(\alpha, \mathbf{r})) + \beta_k h'(\alpha, r)r],
\end{aligned} \tag{13}$$

provided that $f_\chi$ represents the encoding, sampling ad transforming part of the architecture, $g_\psi$ represents the decoding part of the architecture, and $\beta_k, \alpha_k, c_k$ are the parameters of the different transformations. Other types of transformations have been proposed lately. The Householder flow (Tomczak & Welling, 2016) is a volume preserving transformation, meaning that its log determinant equals 1, with the consequence that it can be used with no modifications of the loss function. A more convoluted type of transformations based on a masked autoregressive auto-encoder, the Inverse Autoregressive Flow, was proposed in Kingma & Welling (2013). We did not explore those two last approaches.

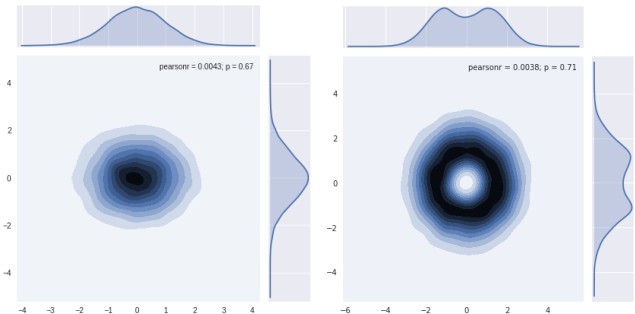

Figure 5: Effect of a Radial Flow transformation on an Isotropic Gaussian Distribution.

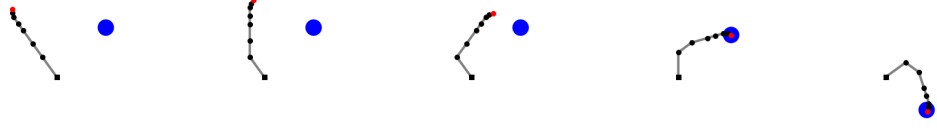

Figure 6: A DMP executed on the Arm-Ball environment.

## C    EXPERIMENTAL ENVIRONMENTS

The following environments were considered:

- **Arm-Ball:** A 7 joints arm, controlled in angular position, can move around in an environment containing a ball. The environment state is perceived visually as a 50x50 pixels image. The arm has a sticky arm tip: if the tip of the arm touches the ball, the ball sticks to the arm until the end of the movement. The underlying state of the environment is hence parameterized by two bounded continuous factors which represent the coordinates of the ball. A situation can be sampled by the experimenter by taking a random point in $[0, 1]^2$.

- **Arm-Arrow:** The same arm can manipulate an arrow in a plane, an arrow being considered as an object with a single symmetry that can be oriented in space. Consequently, the underlying state of the environment is parameterized by two bounded continuous factors representing the coordinates of the arrow , and one periodic continuous factor representing its orientation. A particular situation can hence be sampled by taking a random point in $[0, 1]^3$.

The physical situations were represented by small 70x70 images very similar to the *dSprites* dataset proposed by Higgins et al. (2016)[15]. The arm was not depicted in the field of view of the (virtual) camera used to gather images for representation learning. We used a robotic arm composed of 7 joints, whose motions were parameterized by DMPs using 3 basis functions (hence action policies have 21 continuous parameters), during 50 time-steps. An example of such a DMP executed in the environment is represented in Figure 6. The first phase, where the learner observes changes of the environment (= ball moves) caused by another agent, is modeled by a process which samples iteratively a random state in the underlying state space, e.g. in the case of Arm-Ball $s \sim \mathcal{U}([0, 1]^2)$, and then generating the corresponding image $x = f(s)$ that is observed by the learner.

## D    ALGORITHMIC IMPLEMENTATION

For the experiments, we instantiated the Algorithmic Architecture 1 into Algorithm 3.

In the text, Algorithm 3 is denoted (RGE-⋆), where ⋆ denotes any representation learning algorithm: (RGE-AE) for Auto-Encoders, (RGE-VAE) for Variational Auto-Encoders, (RGE-RFVAE)

---

[15]Available at `https://github.com/deepmind/dsprites-dataset` .

for Radial Flow Variational Auto-Encoders, (RGE-ISOMAP) for Isomap, (RGE-PCA) for Principal Component Analysis and (RGE-FI) for Full Information.

---

**Algorithm 3:** Random Goal Exploration with Unsupervised Goal Space Learning

---

**Input:**

$k$-neighbors regressor $\tilde{D}$, History $\mathcal{H}$, Meta-Policy $\Pi$ is a tabular minimization over $\mathcal{H}$, Unsupervised representation learning algorithm $\mathcal{A}$ (e.g. AE, VAE, Isomap), Kernel Density Estimator algorithm $\mathcal{KDE}$, Random exploration noise $\gamma_m$, Random exploration ratio $\Gamma_e$

1 **begin**
2     **for** *10000 Observation Iterations* **do**
3        Observe a random environment image $x_i$
4        Add this image to a database $\mathcal{D} = \{x_i\}_{i \in [0,10000]}$
5     Learn an embedding function $\tilde{R} : \; x \to o$ using algorithm $\mathcal{A}$ on data $\mathcal{D}$
6     Estimate the outcome distribution $p_{kde}(o)$ from $\{\tilde{R}(x_i)\}_{i \in [0,10000]}$ using algorithm $\mathcal{KDE}$
7     Set the Goal Policy $\gamma = p_{kde}$ to be the estimated outcome distribution
8     **for** *100 Bootstrapping iterations* **do**
9        Sample a random parameterization $\theta_i \sim p(\theta)$
10        Execute the experiment $\theta_i$ (= run a controller with parameters $\theta_i$)
11        Retrieve the outcome from raw image $o_i = \tilde{R}(x_i)$
12        Update the forward model with $\tilde{D}(\theta_i) \triangleq o_i$
13        $\mathcal{H} = \mathcal{H} \cup \{\theta, o\}$
14     **for** *5000 Exploration iterations* **do**
15        **if** $u \sim \mathcal{U}(0,1) < \Gamma_e$ **then**
16           Sample a random parameterization $\theta_i \sim p(\theta)$
17        **else**
18           Sample a goal $g_i \sim \gamma$
19           Sample an exploration noise $\epsilon \sim \mathcal{N}(\mathbf{0}, \mathbf{I})$
20           Execute $\Pi$ to find $\theta_i = \arg\min_{\theta \in \mathcal{H}} C_\tau(\tilde{D_{running}}(\theta))$
21           $\theta_i = \theta_i + \epsilon$
22        Execute the experiment $\theta_i$
23        Retrieve the outcome from raw image $o_i = \tilde{R}(x_i)$
24        Update the forward model with $\tilde{D}(\theta_i) \triangleq o_i$
25        $\mathcal{H} = \mathcal{H} \cup \{\theta, o\}$

26 **return** *The forward model $\tilde{D}$, the history $\mathcal{H}$ and the embedding $\tilde{R}$*

---

## E DETAILS OF NEURAL ARCHITECTURES

Fig. 7 shows the neural networks architectures used for Deep Representation Learning algorithms. Those architectures are based on the one proposed in Higgins et al. (2016).

**Auto-Encoder** The architecture was trained directly without particular stacking. The **AdaGrad** optimizer was used, with initial learning rate of $1e-3$, with batches of size 100, until convergence at $2e5$ epochs.

**Variational Auto-Encoder** The architecture was trained with a *deterministic warm-up* of $1e4$ epochs, as proposed in Sonderby et al. (2016), which shows improved convergence rate. The **Adam** optimizer was used, with initial learning rate of $1e-3$, with batches of size 100, until convergence at $1e5$ epochs.

**Radial Flow Variational Auto-Encoder** The architecture was trained with a *deterministic warm-up* of $1e4$ epochs. The complete flow was made out of 10 planar flows as proposed in Rezende & Mohamed (2015), whose parameters were learned by the encoder. The **Adam** optimizer was used, with initial learning rate of $1e-3$, with batches of size 100, until convergence at $5e4$ epochs.

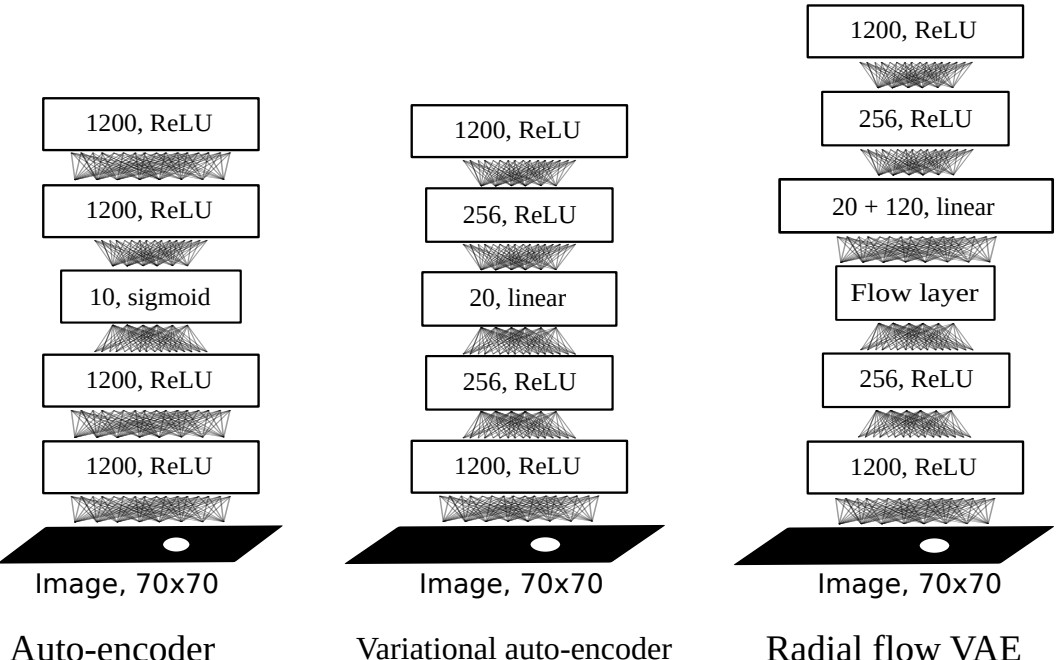

Figure 7: Layers of the different neural networks architectures.

## F EXPLORATION CURVES

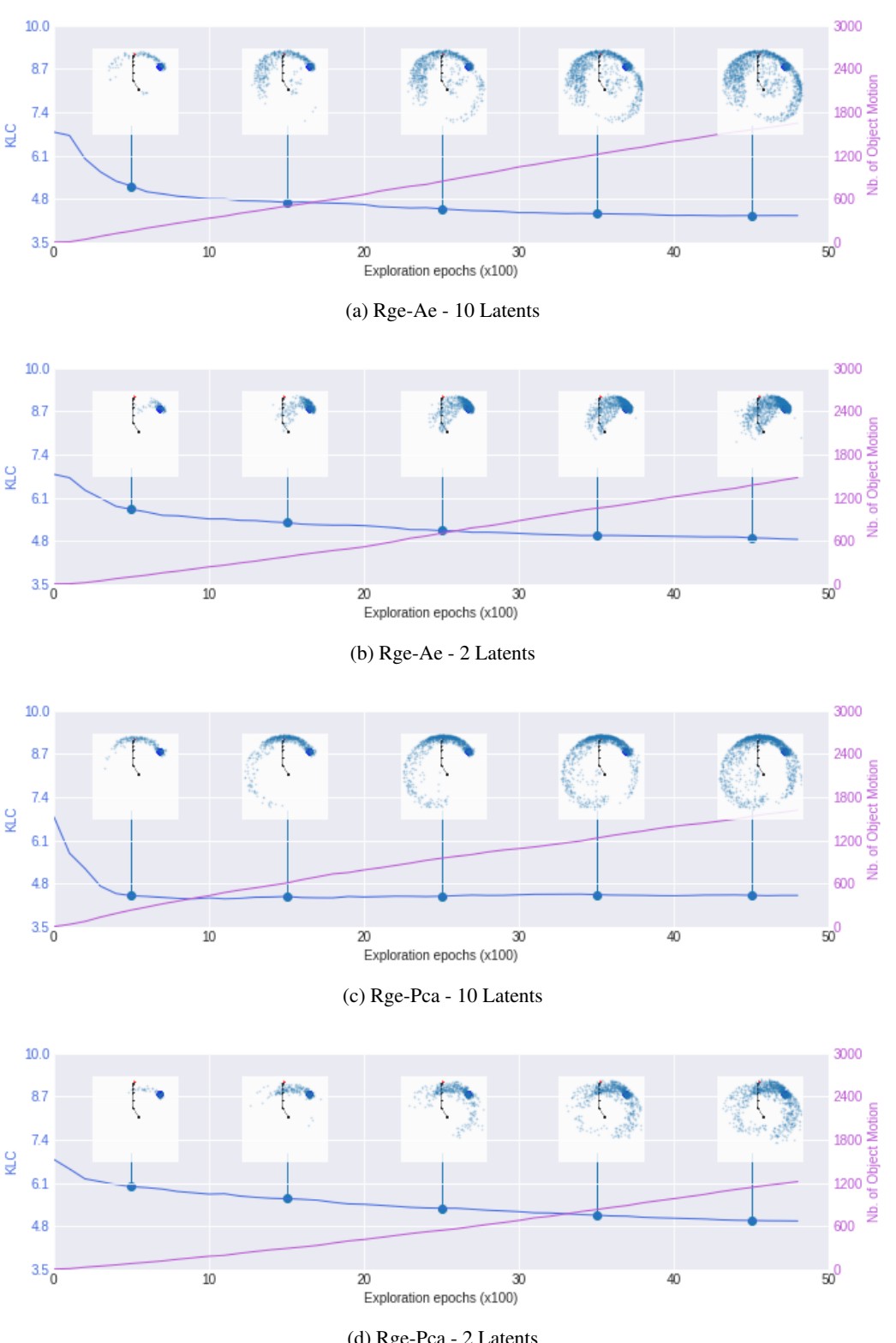

(a) Rge-Ae - 10 Latents

(b) Rge-Ae - 2 Latents

(c) Rge-Pca - 10 Latents

(d) Rge-Pca - 2 Latents

Figure 8: Examples of achieved outcomes related with the evolution of KL-Coverage in the ArmBall environments. The number of times the ball was effectively handled is also represented.

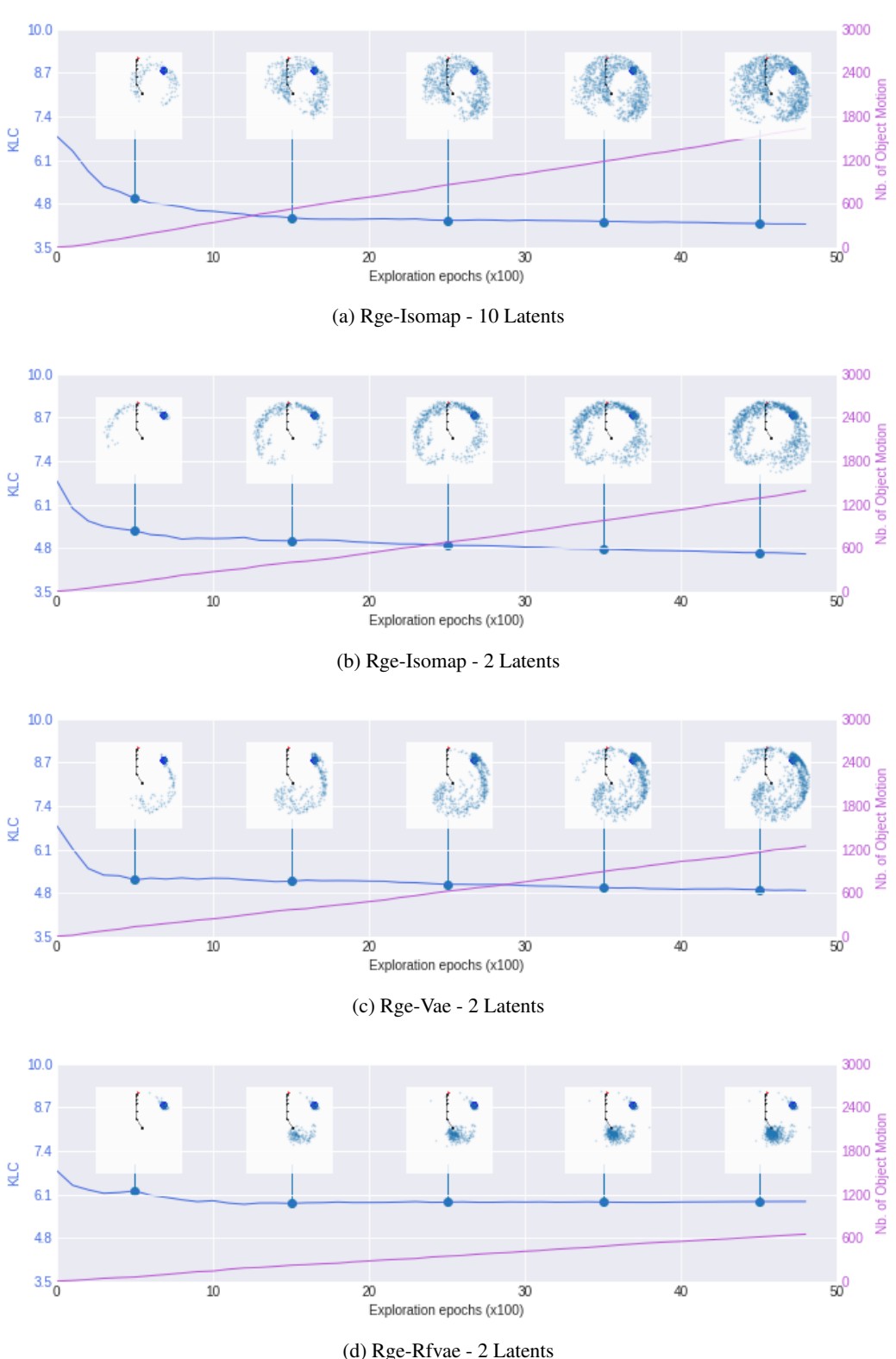

(a) Rge-Isomap - 10 Latents

(b) Rge-Isomap - 2 Latents

(c) Rge-Vae - 2 Latents

(d) Rge-Rfvae - 2 Latents

Figure 9: Examples of achieved outcomes related with the evolution of KL-Coverage in the ArmBall environments. The number of times the ball was effectively handled is also represented.

