# OpenReview forum: "Unsupervised Learning of Goal Spaces for Intrinsically Motivated Goal Exploration"
_ICLR.cc/2018/Conference — Accept (Poster)_

### Official Review · AnonReviewer2 · 2017-11-23
**Review of Unsupervised Learning of Goal Spaces for Intrinsically Motivated Exploration**

**Rating:** 7
**Confidence:** 4

**Review:**

This paper introduces a representation learning step in the Intrinsically Motivated Exploration Process (IMGEP)  framework.

Though this work is far from my expertise fields I find it quite easy to read and a good introduction to IMGEP.
Nevertheless I have some major concerns that prevent me from giving an acceptance decision.

1) The method uses mechanisms than can project back and forth a signal  to the "outcome" space. Nevertheless only the encoder/projection part seems to be used in the algorithm presented p6. For example the encoder part of an AE/VAE is used as a preprocesing stage of the phenomenon dynamic D. It should be obviously noticed that the decoder part could also be used for helping the inverse model I but apparently that is not the case in the proposed method.

2) The representation stage R seems to be learned at the beginning of the algorithm and then fixed. When using DNN as R (when using AE/VAE) why don't you propagate a gradient through R when optimizing D and I ? In this way, learning R at the beginning is only an old good pre-training of DNN with AE.

3) Eventually, Why not directly considering R as lower layers of D and using up to date techniques to train it ? (drop-out, weight clipping, batch normalization ...).
Why not using architecture adapted to images such as CNN ?

---

> ### Author Response · Authors · 2018-01-05
> **Specific responses to reviewer 2**
>
> These comments suggest that the reviewer thinks that in the particular experiment we made, and thus the particular implementation of IMGEPs we used, we are training a single large neural network for learning forward and inversed models. We could have done this indeed, and in that case the reviewer' suggestion would recommend very relevantly to use the lower-layers and/or decoding projection of the (variational) auto-encoders. However, we are not using neural networks for learning forward and inverse models, but rather non-parametric methods based on memorizing examplars associating the parameters of DMPs and their outcomes in the embedding space (which itself comes from auto-encoders),
> in combination with local online regression models and optimization on these local models. This approach comes from the field of robotics, where is has shown extremely efficient for fast incremental learning of forward and inverse models. Comparing this approach with a full neural network approach (which might generalize better but have difficulties for fast incremental learning) would be a great topic for another paper. In the new version of the article, we have tried to improve the clarity of the description of the particular implementation of IMGEPs we have used.

---

### Official Review · AnonReviewer1 · 2017-11-27
**Interesting, but not substantial enough -> updated now good enough**

**Rating:** 7
**Confidence:** 4

**Review:**

The paper investigates different representation learning methods to create a latent space for intrinsic goal generation in guided exploration algorithms.  The research is in principle very important and interesting.

The introduction discusses a great deal about intrinsic motivations and about goal generating algorithms. This is really great, just that the paper only focuses on a very small aspect of learning a state representation in an agent that has no intrinsic motivation other than trying to achieve random goals.
I think the paper (not only the Intro) could be a bit condensed to more concentrate on the actual contribution.

The contribution is that the quality of the representation and the sampling of goals is important for the exploration performance and that classical methods like ISOMap are better than Autoencoder-type methods.

Also, it is written in the Conclusions (and in other places): "[..] we propose a new intrinsically Motivated goal exploration strategy....". This is not really true.  There is nothing new with the intrinsically motivated selection of goals here, just that they are in another space. Also, there is no intrinsic motivation. I also think the title is misleading.

The paper is in principle interesting. However, I doubt that the experimental evaluations are substantial enough for profound conclusion.

Several points of critic:
- the input space was very simple in all experiments, not suitable for distinguishing between the algorithms, for instance, ISOMap typically suffers from noise and higher dimensional manifolds, etc.
- only the ball/arrow was in the input image, not the robotic arm. I understand this because in phase 1 the robot would not move, but this connects to the next point:
- The representation learning is only a preprocessing step requiring a magic first phase.
    -> Representation is not updated during exploration
- The performance of any algorithm (except FI) in the Arm-Arrow task is really bad but without comment.
- I am skeptical about the VAE  and RFVAE results. The difference between Gaussian sampling and the KDE is a bit alarming, as the KL in the VAE training is supposed to match the p(z) with N(0,1). Given the power of the encoder/decoder it should be possible to properly represent the simple embedded 2D/3D manifold and not just a very small part of it as suggested by Fig 10.
I have a hard time believing these results. I urge you to check for any potential errors made. If there are not mistakes then this is indeed alarming.

Questions:
- Is it true that the robot always starts from same initial condition?! Context=Emptyset.
- For ISOMap etc, you also used a 10dim embedding?

Suggestion:
- The main problem seems to be that some algorithms are not representing the whole input space.
- an additional measure that quantifies the difference between true input distribution and reproduced input distribution could tier the algorithms apart and would measure more what seems to be relevant here.  One could for instance measure the KL-divergence between the true input and the sampled (reconstructed) input (using samples and KDE or the like).
- This could be evaluated on many different inputs (also those with a bit more complicated structure) without actually performing the goal finding.
- BTW: I think Fig 10 is rather illustrative and should be somehow in the main part of the paper

On the positive side, the paper provides lots of details in the Appendix.
Also, it uses many different Representation Learning algorithms and uses measures from manifold learning to access their quality.

In the related literature, in particular concerning the intrinsic motivation, I think the following papers are relevant:
J. Schmidhuber, PowerPlay: training an increasingly general problem solver by continually searching for the simplest still unsolvable problem. Front. Psychol., 2013.

and

G. Martius, R. Der, and N. Ay. Information driven self-organization of complex robotic behaviors. PLoS ONE, 8(5):e63400, 2013.


Typos and small details:
p3 par2: for PCA you cited Bishop. Not critical, but either cite one the original papers or maybe remove the cite altogether
p4 par-2: has multiple interests...: interests -> purposes?
p4 par-1: Outcome Space to the agent is is ...
Sec 2.2 par1: are rapidly mentioned... -> briefly
Sec 2.3 ...Outcome Space O, we can rewrite the architecture as:
  and then comes the algorithm. This is a bit weird
Sec 3: par1: experimental campaign -> experiments?
p7: Context Space: the object was reset to a random position or always to the same position?
Footnote 14: superior to -> larger than
p8 par2: Exploration Ratio Ratio_expl... probably also want to add (ER) as it is later used
Sec 4: slightly underneath -> slightly below
p9 par1: unfinished sentence: It is worth noting that the....
one sentence later: RP architecture? RPE?
Fig 3: the error of the methods (except FI) are really bad. An MSE of 1 means hardly any performance!
p11 par2: for e.g. with the SAGG..... grammar?

Plots in general: use bigger font sizes.

---

> ### Author Response · Authors · 2018-01-05
> **Specific responses to reviewer 1 (part 1)**
>
> > R1 "an agent that has no intrinsic motivation other than trying to achieve random goals."
> "There is nothing new with the intrinsically motivated selection of goals here, just that they are in another space. Also, there is no intrinsic motivation. I also think the title is misleading."
>
> The concept of "intrinsically motivated learning and exploration" is not yet completely well-defined across (even computionational) communities, and we agree that the use of the term "intrinsically motivated exploration" in this article may seem unusual for some readers. However, we strongly think it makes sense to keep it for the following reasons.
>
> There are several conceptual approaches to the idea of "intrinsically motivated learning and exploration", and we believe our use of the term intrinsic-motivation is compatible with all of them:
>
> - Focus on task-independance and self-generated goals: one approach of intrinsic motivation, rooted in its conceptual origins in psychology, is that it designates the set of mechanisms and behaviours of organized exploration which are not directed towards a single extrinsically imposed goal/problem (or towards fullfilling physiological motivations like food search), but rather are self-organized towards intrinsically defined objectives and goals (independant of physiological motivations like food search). From this perspective, mechanisms that self-generate goals, even randomly, are maybe the simplest and most prototypical form of intrinsically motivated exploration.
>
> - Focus on information-gain or competence-gain driven exploration: Other approaches consider that intrinsically motivated exploration specifically refers to mechanisms where choices of actions or goals are based on explicit measures of expected information-gain about a predictive model, or novelty or surprise of visited states, or competence gain for self-generated goals.  In the IMGEP framework, this corresponds specifically to IMGEP implementations where the goal sampling procedure is not random, but rather based on explicit estimations of expected competence gain, like in the SAGG-RIAC architecture or in modular IMGEPs of (Forestier et al., 2017). In the experiments presented in this article, the choice of goals is made randomly as the focus is not on the efficiency of the goal sampling policy. However, it would be straightforward to use a selection of goals based on expected competence gain, and thus from this perspective the proposed algorithm adresses the general problem of how to learn goal representations in IMGEPs.
>
> - Focus on noverly/diversity search mechanisms: Yet another approach to intrinsically motivated learning and exploration is one that refers to mechanisms that organize the learner's exploration so that exploration of novel or diverse behaviours is fostered. A difference with the previous approach is that here one does not necessarily use internally a measure of novelty or diversity, but rather one uses it to characterize the dynamics of the behaviour. And an interesting property of random goal exploration implementations of IMGEPs is that while it does not measure explicitly novelty or diversity, it does in fact maximize it through the following mechanism: from the beginning and up to the point where the a large proportion of the space has been discovered, generating random goals will very often produce goals that are outside the convex hull of already discovered goals. This in turn mechanically leads to exploration of stochastic variants of motor programs that produce outcomes on the convex hull, which statistically pushes the convex hull further, and thus fosters exploration of motor programs that have a high probability to produce novel outcomes outside the already known convex hull.

---

> > ### Author Response · Authors · 2018-01-05
> > **Specific responses to reviewer 1 (part 2)**
> >
> > > R1 "The representation learning is only a preprocessing step requiring a magic first phase.
> > >    -> Representation is not updated during exploration"
> > > "- only the ball/arrow was in the input image, not the robotic arm. I understand this because in phase 1 the robot would not move, but this connects to the next point:"
> >
> > Indeed, representation is not updated during exploration, and as mentioned in the conclusion we think doing this is a very important direction for future work. However, we have two strong justification for this decomposition, that we added in the paper.
> >
> > First, we do not believe the preliminary pre-processing step is "magical". Indeed, if one studies the work from the developmental learning perspective outlined in the introduction, where one takes inspiration from the processes of learning in infants, then this decomposition corresponds to a well-known developmental progression: in their first few weeks, motor exploration in infants is very limited (due to multiple factors), while they spend a considerable amount of time observing what is happening in the outside world with their eyes (e.g. observing images of others producing varieties of effects on objects). During this phase, a lot of perceptual learning happens, and this is reused later on for motor learning (infant perceptual development often happens ahead of motor development in several important ways). In the article, the concept of "social guidance" presented in the introduction, and the availability of a database of observations of visual effects that can happen in the world, can be seen as a model of this first phase of infant learning by passively observing what is happening around them.
> >
> > A second justification for this decomposition is more methodological. It is mainly an experimental tool for better understanding what is happening. Indeed, the underlying algorithmic mechanisms are already quite complex, and analyzing what is happening when one decomposes learning in these two phases (representation learning, then exploration) is an important scientific step. Presenting in the same article another study where representations would be updated continuously would result in too much material to be clearly presented in a conference paper.
> >
> > > R1 "the input space was very simple in all experiments, not suitable for distinguishing between the algorithms, for instance, ISOMap typically suffers from noise and higher dimensional manifolds"
> >
> > The use of the term "simple" depends on the perspective. From the perspective of a classical goal exploration process that would use the 4900 raw pixels as input, not knowing they are pixels and considering them similarly as when engineered representations are provided, then this is a complicated space and exploration is very difficult. At the same time, from the point of view of representation learning algorithms, this is indeed a moderately complex input space (yet, we on purpose did not consider convolutionnal auto-encoders so that the task is not too simplified and results could apply to other modalities such as sound or proprioception). Third, if one considers the dimensionality of the real sensorimotor manifold in which action is happening (2 for arm-ball, 3 for arm-arrow), this does not seem to us to be too unrealistic as many of real world sensorimotor tasks are actually happening in low-dimensional task spaces (e.g. rigid object manipulation happens in a 6D task space). So, overall we have chosen these experimental setups as we belive they are a good compromise between simplicity (enabling us to understand well what is happening) and complexity (if one considers the learner does not already knows that the stimuli are pixels of an image).

---

> > > ### Author Response · Authors · 2018-01-05
> > > **Specific responses to reviewer 1 (part 3)**
> > >
> > > > R1 "The performance of any algorithm (except FI) in the Arm-Arrow task is really bad but without comment."
> > > See general answer and new graphs in the paper: most algorithms actually perform very well from the main perspective of interest in the paper (exploration efficiency).
> > >
> > > > R1 "- I am skeptical about the VAE  and RFVAE results. If there are not mistakes then this is indeed alarming."
> > > > R1 "- The main problem seems to be that some algorithms are not representing the whole input space.
> > >
> > > Following your remark, we double checked the code and made an in depth verification of results. A small bug indeed existed, which made the projection of points in latent space wider than it should be. This was fixed in those new experiments, and we validated that the whole input space was represented in the latent representation. Despite this, it didn't changed the conclusion drawn in the original paper. Indeed, our new results show the same type of behavior as in the first version, in particular:
> > > 	+ The exploration performances for VAE with KDE goal sampling distribution are still above Gaussian goal Sampling. Our experiments showed that convergence on the KL term of the loss can be more or less quick depending on the initialization. Since we used an number of iterations as stopping criterion for our trainings (based on early experiments), we found that sometimes, at stop, despite achieving a low reconstruction error, the divergence was still pretty high. In those cases the representation was not perfectly matching an isotropic gaussian, which lead to biased sampling.
> > >     + The performances of the RFVAE are still worse than any other algorithms. Our experiments showed that they introduce a lot of discontinuities in the representation, which along with physics boundaries of achievable states, can generate "pockets" in the representation from which a Random Goal Exploration can't escape. This would likely be different for a more advanced exploration strategy such as Active Goal exploration.
> > >
> > > > R1 - Is it true that the robot always starts from same initial condition?! Context=Emptyset.
> > >
> > > yes. In (Forestier et al., ICDL-Epirob 2016), a similar setup is used except that the starting conditions are randomized at each new episode (and that goal representation are engineered): they show that the dynamics of exploration scales wells. Here we chose to start from the same initial condition to be able to display clearly in 2D the full space of discovered outcomes (if one would include the starting ball position, this would be a 4D space).
> > >
> > > > R1 - For ISOMap etc, you also used a 10dim embedding?
> > >
> > > yes.
> > >
> > > >In the related literature, in particular concerning the intrinsic motivation, I think the following papers are relevant:
> > > >J. Schmidhuber, PowerPlay: training an increasingly general problem solver by continually searching for the simplest still unsolvable problem. Front. Psychol., 2013.
> > > >and
> > > >G. Martius, R. Der, and N. Ay. Information driven self-organization of complex robotic behaviors. PLoS ONE, 8(5):e63400, 2013.
> > >
> > > yes, these are relevant papers indeed, which are cited in reviews we cite, but we added them for more coverage.

---

### Official Review · AnonReviewer3 · 2017-11-29
**Some interesting ideas, yet no clear message**

**Rating:** 6
**Confidence:** 2

**Review:**

[Edit: After revisions, the authors have made a good-faith effort to improve the clarity and presentation of their paper: figures have been revised, key descriptions have been added, and (perhaps most critically) a couple of small sections outlining the contributions and significance of this work have been written. In light of these changes, I've updated my score.]

Summary:

The authors aim to overcome one of the central limitations of intrinsically motivated goal exploration algorithms by learning a representation without relying on a "designer" to manually specify the space of possible goals. This work is significant as it would allow one to learn a policy in complex environments even in the absence of a such a designer or even a clear notion of what would constitute a "good" distribution of goal states.

However, even after multiple reads, much of the remainder of the paper remains unclear. Many important details, including the metrics by which the authors evaluate performance of their work, can only be found in the appendix; this makes the paper very difficult to follow.

There are too many metrics and too few conclusions for this paper. The authors introduce a handful of metrics for evaluating the performance of their approach; I am unfamiliar with a couple of these metrics and there is not much exposition justifying their significance and inclusion in the paper. Furthermore, there are myriad plots showing the performance of the different algorithms, but very little explanation of the importance of the results. For instance, in the middle of page 9, it is noted that some of the techniques "yield almost as low performance as" the randomized baseline, yet no attempt is made to explain why this might be the case or what implications it has for the authors' approach. This problem pervades the paper: many metrics are introduced for how we might want to evaluate these techniques, yet there is no provided reason to prefer one over another (or even why we might want to prefer them over the classical techniques).

Other comments:
- There remain open questions about the quality of the MSE numbers; there are a number of instances in which the authors cite that the "Meta-Policy MSE is not a simple to interpret" (The remainder of this sentence is incomplete in the paper), yet little is done to further justify why it was used here, or why many of the deep representation techniques do not perform very well.
- The authors do not list how many observations they are given before the deep representations are learned. Why is this? Additionally, is it possible that not enough data was provided?
- The authors assert that 10 dimensions was chosen arbitrarily for the size of the latent space, but this seems like a hugely important choice of parameter. What would happen if a dimension of 2 were chosen? Would the performance of the deep representation models improve? Would their performance rival that of RGE-FI?
- The authors should motivate the algorithm on page 6 in words before simply inserting it into the body of the text. It would improve the clarity of the paper.
- The authors need to be clearer about their notation in a number of places. For instance, they use \gamma to represent the distribution of goals, yet it does not appear on page 7, in the experimental setup.
- It is never explicitly mentioned exactly how the deep representation learning methods will be used. It is pretty clear to those who are familiar with the techniques that the latent space is what will be used, but a few equations would be instructive (and would make the paper more self-contained).

In short, the paper has some interesting ideas, yet lacks a clear takeaway message. Instead, it contains a large number of metrics and computes them for a host of different possible variations of the proposed techniques, and does not include significant explanation for the results. Even given my lack of expertise in this subject, the paper has some clear flaws that need addressing.

Pros:
- A clear, well-written abstract and introduction
- While I am not experienced enough in the field to really comment on the originality, it does seem that the approach the authors have taken is original, and applies deep learning techniques to avoid having to custom-design a "feature space" for their particular family of problems.

Cons:
- The figure captions are all very "matter-of-fact" and, while they explain what each figure shows, provide no explanation of the results. The figure captions should be as self-contained as possible (I should be able to understand the figures and the implications of the results from the captions alone).
- There is not much significance in the current form of the paper, owing to the lack of clear message. While the overarching problem is potentially interesting, the authors seem to make very little effort to draw conclusions from their results. I.e. it is difficult for me to easily visualize all of the "moving parts" of this work: a figure showing the relationship bet
- Too many individual ideas are presented in the paper, hurting clarity. As a result, the paper feels scattered. The authors do not have a clear message that neatly ties the results together.

---

> ### Author Response · Authors · 2018-01-05
> **Specific responses to reviewer 3**
>
> > R3 "does not include significant explanation for the results", "The figure captions are all very "matter-of-fact" and, while they explain what each figure shows, provide no explanation of the results."
> We agree. We have added several more detailed explanations of the results.
>
> > R3 "why many of the deep representation techniques do not perform very well."
> We think this comment is due to our unclear explanation of our main target combined with the use of a misleading measure (MSE). We hope the new explanation we provide, as well as the focus on exploration measures based on the KL divergence will enable to make it more clear that on the contrary several deep learning approaches are performing very well, some systematically outperforming the use of handcrafted goal space features (see the common answer to all reviewers).
>
> > R3 "The authors assert that 10 dimensions was chosen arbitrarily for the size of the latent space, but this seems like a hugely important choice of parameter. What would happen if a dimension of 2 were chosen? Would the performance of the deep representation models improve? Would their performance rival that of RGE-FI?"
>
> We agree that this is a very important point. We have in the new version included results when one gives algorithms the right number of dimensions (2 for arm-ball, 3 for arm-arrow), and showing that providing more dimensions to IMGEP-UGL algorithms than the "true" dimensionality of the phenomenon can actually be beneficial (and we provide an explanation why this is the case).
>
> > "The authors do not list how many observations they are given before the deep representations are learned. Why is this? Additionally, is it possible that not enough data was provided?"
>
> For each environments, we trained the networks with a dataset of 10.000 elements uniformly sampled in the underlying state-space. This corresponds to 100 samples per dimension for the 'armball' environment, and around 20 per dimension for the 'armarrow' environment. This is not far from the number of samples considered in the dsprite dataset, in which around 30 samples per dimensions are considered. Moreover, our early experiments showed that for those two particular problems, adding more data did not change the exploration results.
>
> > "- The authors should motivate the algorithm on page 6 in words before simply inserting it into the body of the text. It would improve the clarity of the paper."
>
> We have tried to better explain in words the general principles of this algorithm.
>
> > "The authors need to be clearer about their notation in a number of places. For instance, they use gamma to represent the distribution of goals, yet it does not appear on page 7, in the experimental setup."
>
> We have tried to correct these problems in notations.
>
> > "It is never explicitly mentioned exactly how the deep representation learning methods will be used. It is pretty clear to those who are familiar with the techniques that the latent space is what will be used, but a few equations would be instructive (and would make the paper more self-contained)."
>
> yes indeed. We have added some new explanations.

---

### Author Response · Authors · 2018-01-05
**General answer to all reviewers**

We thank all reviewers for their detailed comments, which have helped us a lot to improve our paper. On one hand, we appreciate that all reviewers found the overall approach interesting and important.
On the other hand, we agree with reviewers that there were shortcomings in paper, and we thank them for pointing ways in which it could be improved, which we have attempted to do in the new version of the article, that includes both new explanations and new experimental results.

The main point of the reviewers was that our text did not identify concisely and clearly the main contributions and conclusions of this article, and in particular did not enable the reader to rank the importance and focus of these contributions (from our point of view). The comment of reviewer R1, summarizing our contributions, actually shows that we have not explained clearly enough what was our main target contribution (see below).
We have added an explicit paragraph at the end of the introduction to outline and rank our contributions, as well as a paragraph at the beginning of the experimental section to pin point the specific questions to which the experiments provide an answer. We hope the messages are now much clearer.

Another point was that our initial text contained too many metrics, and lacked justification of their choices and relative importance. We have rewritten the results sections by focusing in more depth on the most important metrics (related to our target contributions), updating some of them with more standard metrics, and removing some more side metrics. The central property we are interested in in this article is the dynamics and quality of exploration of the outcome space, characterizing  the (evolution of the) distribution of discovered outcomes, i.e. the diversity of effects that the learner discovers how to produce. In the initial version of the article, we used an ad hoc measure called "exploration ratio" to characterize the evolution of the global quality of exploration of an algorithm. We have now replaced this ad hoc measure with a more principled and more precise measure: the KL divergence between the discovered distribution of outcomes and the distribution produced by an oracle (= uniform distribution of points over the reachable part of the outcome space). This new measure is more precise as it much better takes into account the set of roll-outs which do not make the ball/arrow move at all. In the new version of the article, we can now see that this more precise measure enables to show that several algorithms actually approximate extremely well the dynamics of exploration IMGEPs using a goal space with engineered features, and that even some IMGEP-UGL algorithms (RGE-VAE) systematically outperform this baseline algorithm. Furthermore, we have now included plots of the evolution of the distribution of discovered outcomes in individual runs to enable the reader to grasp more clearly the progressive exploration dynamics for each algorithms.

Another point was that the MSE measure used in the first version of the article was very misleading. Indeed, it did not evaluate the exploration dynamics, but rather it evaluated a peculiar way to reuse in combination both the discovered data points and the learned representation in a particular kind of test (raw target images were given to the learner). This was misleading because 1) we did not explain well that it was evaluating this as opposed to the main target of this article (distribution of outcomes); 2) this test evaluates a rather exotic way to reuse the discovered data points (previous papers reused the discovered data in other ways). This lead R1 to infer that the algorithms were not not working well in comparison with the “Full Information” (FI) baseline (now called EFR, for "Engineered Feature Representation"): on the contrary, several IMGEP-UGL algorithms actually perform better from the perspective we are interested in here. As the goal of this paper is not to study how the discovered outcomes can be reused for other tasks, we have removed the MSE measures.

---

### Decision · Program_Chairs · 2018-01-29
**ICLR 2018 Conference Acceptance Decision**

**Decision:**

Accept (Poster)

**Comment:**

This paper aims to improve on the intrinsically motivated goal exploration framework by additionally incorporating representation learning for the space of goals. The paper is well motivated and follows a significant direction of research, as agreed by all reviewers. In particular, it provides a means for learning in complex environments, where manually designed goal spaces would not be available in practice. There had been significant concerns over the presentation of the paper, but the authors put great effort in improving the manuscript according to the reviewers’ suggestions, raising the average rating by 2 points after the rebuttal.